# Optimizing Automatic Differentiation with Deep Reinforcement Learning

**Jamie Lohoff**
Peter Grünberg Institute
Forschungszentrum Jülich & RWTH Aachen
`ja.lohoff@fz-juelich.de`

**Emre Neftci**
Peter Grünberg Institute
Forschungszentrum Jülich & RWTH Aachen
`e.neftci@fz-juelich.de`

## Abstract

Computing Jacobians with automatic differentiation is ubiquitous in many scientific domains such as machine learning, computational fluid dynamics, robotics, and finance. Even small savings in the number of computations or memory usage in Jacobian computations can already incur massive savings in energy consumption and runtime. While there exist many methods that allow for such savings, they generally trade computational efficiency for approximations of the exact Jacobian. In this paper, we present a novel method to optimize the number of necessary multiplications for Jacobian computation by leveraging deep reinforcement learning (RL) and a concept called cross-country elimination while still computing the exact Jacobian. Cross-country elimination is a framework for automatic differentiation that phrases Jacobian accumulation as ordered elimination of all vertices on the computational graph where every elimination incurs a certain computational cost. We formulate the search for the optimal elimination order that minimizes the number of necessary multiplications as a single player game which is played by an RL agent. We demonstrate that this method achieves up to 33% improvements over state-of-the-art methods on several relevant tasks taken from diverse domains. Furthermore, we show that these theoretical gains translate into actual runtime improvements by providing a cross-country elimination interpreter in JAX that can efficiently execute the obtained elimination orders.

## 1   Introduction

Automatic Differentiation (AD) is widely utilized for computing gradients and Jacobians across diverse domains including machine learning (ML), computational fluid dynamics (CFD), robotics, differential rendering, and finance [Baydin et al., 2018, Margossian, 2018, Forth et al., 2004a, Tadjouddine et al., 2002b, Giftthaler et al., 2017, Kato et al., 2020, Schmidt et al., 2022, Capriotti and Giles, 2011, Savine and Andreasen, 2021]. To many researchers in the machine learning community, AD is synonymous with the backpropagation algorithm [Linnainmaa, 1976, Schmidhuber, 2014]. However, backpropagation is just one particular way of algorithmically computing the Jacobian that is very efficient in terms of computations for "funnel-like" functions, *i.e.* with many inputs and a single scalar output such as in neural networks. In many other domains, we may find functions that do not have this particular property and thus backpropagation might not be optimal for computing the respective Jacobian [Albrecht et al., 2003, Capriotti and Giles, 2011, Naumann, 2020]. In fact, there exists a wide variety of AD algorithms, each of them coming with its own advantages and drawbacks regarding computational cost and memory consumption depending on the function they are applied to. Many of these AD algorithms can be viewed as special cases of *cross-country elimination* [Griewank and Walther, 2008]. Cross-County Elimination frames AD as an ordered vertex elimination problem on the computational graph with the goal of reducing the required number of multiplications and additions. However, finding the optimal elimination procedure is a NP-complete problem [Naumann,

38th Conference on Neural Information Processing Systems (NeurIPS 2024).

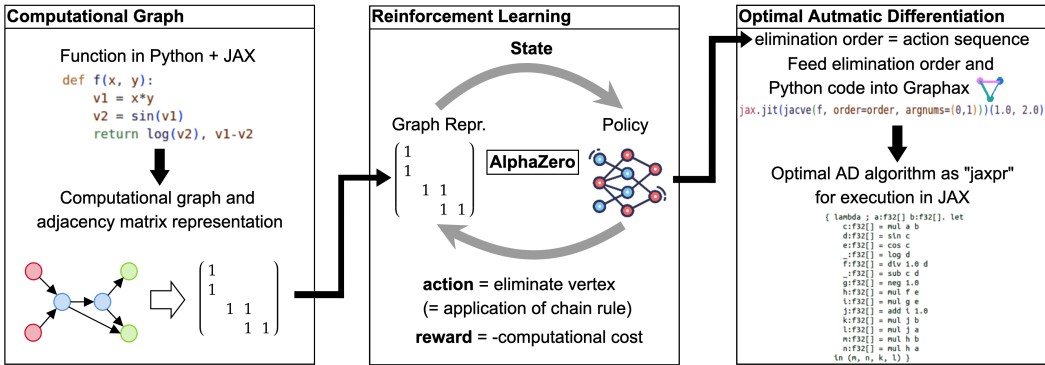

Figure 1: Summary of the AlphaGrad pipeline. We trained a neural network to produce new Automatic Differentiation (AD) algorithms using Deep RL that can be used in JAX. The resulting algorithms significantly outperform the current state of the art.

2008]. Inspired by recent advances in finding optimal matrix-multiplication and sorting algorithms [Fawzi et al., 2022, Mankowitz et al., 2023], we demonstrate that deep RL successfully finds efficient elimination orders which translate into new automatic differentiation algorithms and practical runtime gains (figure 1). Cross-country elimination is particularly amenable for automatization since it provably yields the exact Jacobian for every elimination order. The solution we seek thus reduces to only finding an optimal elimination order, without the need to evaluate the quality of Jacobian approximations (such as in Neural Architecture Search). An important body of prior work aimed to find more efficient elimination techniques through heuristics, simulated annealing or dynamic programming and minimizing related quantities such as fill-in [Naumann, 1999, 2020]. However, none of these works was successful in optimizing with respect to relevant quantities such as number of multiplications or memory consumption. We set up our optimization problem by formulating cross-country elimination as a single player RL game called *VertexGame*. At each step of VertexGame, the agent selects a vertex to eliminate from the computational graph according to a certain scheme called *vertex elimination*. The reward is equal to the negative of the number of multiplications incurred by the particular choice of vertex. VertexGame is played by an AlphaZero-based agent [Silver et al., 2017, Schrittwieser et al., 2019, Danihelka et al., 2022] with policy and value functions modeled with a transformer architecture that processes the graph representation and predicts the next vertex to eliminate, thereby incrementally building the AD algorithm.

Our approach discovers from scratch new vertex elimination orders, *i.e.* new AD algorithms that are tailored to specific functions and improve over the established methods such as minimal Markowitz degree. We further demonstrate the efficacy of the discovered algorithms on real world tasks by including *Graphax*, a novel sparse AD package which builds on JAX [Bradbury et al., 2018] and enables the user to differentiate Python code with cross-country elimination. Our main contributions are summarized as follows:

- We demonstrate that optimizing elimination order can be phrased as a reinforcement learning game by leveraging the graph view of AD,

- We show that a deep RL agent finds new, tailored AD algorithms that improve the state-of-the-art on several relevant tasks,

- We investigate how the discovered novel elimination procedures translate into actual runtime improvements by implementing Graphax, a cross-country elimination interpreter in JAX allowing the efficient execution of newly found elimination orders.

## 1.1 Related Work

**RL for Algorithm Research:** AlphaTensor and AlphaDev successfully demonstrated that model-based deep RL finds new and improved matrix-multiplication and sorting algorithms [Fawzi et al., 2022, Mankowitz et al., 2023]. In particular, AlphaTensor used an extension of the AlphaZero agent to search for new matrix-multiplications algorithms that require fewer multiplication operations by directly using this quantity as a reward. The key insight is that different matrix-multiplication

algorithms have a common, simple representation through the three-dimensional matrix-multiplication tensor which can be manipulated by taking different actions, resulting in algorithms of varying efficiency. Feeding this tensor into the RL agent, they successfully improved on matrix-multiplication algorithms for 4x4 matrices by beating Strassen's algorithm, the current state-of-the-art, with an improvement from 49 to 47 multiplications.

In a similar vein, AlphaDev improved simple sorting algorithms by representing the sorting algorithm as a series of CPU instructions which then have to be arranged in the correct way to achieve the correct sorting output. Instead of using the number of CPU operations as an optimization target, the agent was trained on actual execution times. Our work follows in these footsteps by tackling the difficult problem of finding new and improved AD algorithms for arbitrary functions and hence we termed our method *AlphaGrad*.

**RL for Compiler Optimization:** A number of works have tackled the complex issue of optimizing the compilation of various computational graphs with deep RL. Knossos [Jinnai et al., 2019] leverages the A$^*$ algorithm to optimize the compilation of simple neural networks. It employs a model to estimate computational cost and utilizes expression rewriting techniques to enhance performance. While Knossos is hardware agnostic, it needs to be trained from scratch for every new computational graph. GO and REGAL both improve on this shortcoming and generalize to new, unseen graphs at the cost of losing the hardware-agnostic property[Paliwal et al., 2019, Zhou et al., 2020].

REGAL learns a graph neural network-based policy using a REINFORCE-based genetic algorithm to optimize the scheduling of the individual operations of a graph to the set of available devices, thereby successfully reducing peak memory usage for different deep learning workloads. Only GO is directly trained on actual wall time and handles all relevant optimizations jointly, including device placement, operation fusion, and operation scheduling. GO learns a policy based on graph neural networks and recurrent attention using PPO and successfully demonstrates improvements over Tensorflow's default compilation strategy. While our work also makes use of the computational graph, the goal is to find novel AD algorithms instead of optimizing compilation itself, although these problems are related since the new AD algorithm is compiled before execution.

**Optimization of AD** While no prior work directly aims at improving AD with deep RL, several studies aimed at enhancing AD by other methods. This includes Enzyme [Moses and Churavy, 2020], which presents a reverse-mode AD package that operates on the intermediate representation level using LLVM. Enzyme is thus distinct from other AD packages because it can synthesize gradients for many different high-level languages. Another related work is LAGrad [Peng and Dubach, 2023], a source-to-source AD package written in Julia which introduces a set of new static optimizations to accelerate reverse-mode AD. It leverages high-level MLIR information, such as sparsity structure and control flow semantics of the computational graph to produce more efficient differentiated code. While LAGrad can improve the performance of AD workloads by orders of magnitude, it is currently limited to the use of reverse-mode AD which can be suboptimal for certain tasks. In both works, a suboptimal choice of algorithm might nullify the benefits gained from careful engineering. Our work aims to close this gap by additionally providing a novel way of finding the optimal AD algorithm using Deep RL.

The closest related work is [Naumann, 1999] where simulated annealing was applied to reduce the number of multiplications necessary for Jacobian accumulation. The algorithms struggled to significantly outperform state-of-the-art even when it was initialized with a reasonably good elimination order. In a similar manner, [Naumann, 2020] directly optimized the elimination order with dynamic programming albeit with respect to a different optimization target called fill-in on randomly generated graphs that do not necessary represent well-defined, executable functions.t Our work directly optimizes for the number of multiplications required to accumulate the Jacobian on real-world problems. Another approach described in [Chen et al., 2012] utilized integer linear programming to find optimal elimination orders with respect to number of multiplications, but only dealt with very small problems with up to twenty intermediate vertices. Our approach successfully finds new AD algorithms from scratch for complex problems with hundreds of intermediate vertices.

## 2 Automatic Differentiation and Cross-Country Elimination

AD is a systematic approach to computing the derivatives of *dependent variables* $\mathbf{y} = f(\mathbf{x}) \in \mathbb{R}^m$ with respect to the *independent variables* $\mathbf{x} \in \mathbb{R}^n$ utilizing the chain rule. AD enables the precise and efficient calculation of gradients, Jacobians, Hessians, and higher-order derivatives [Linnainmaa, 1976]. Unlike methods that rely on finite differences or symbolic differentiation, AD offers a systematic way to compute derivatives up to machine precision, making it an indispensable tool in

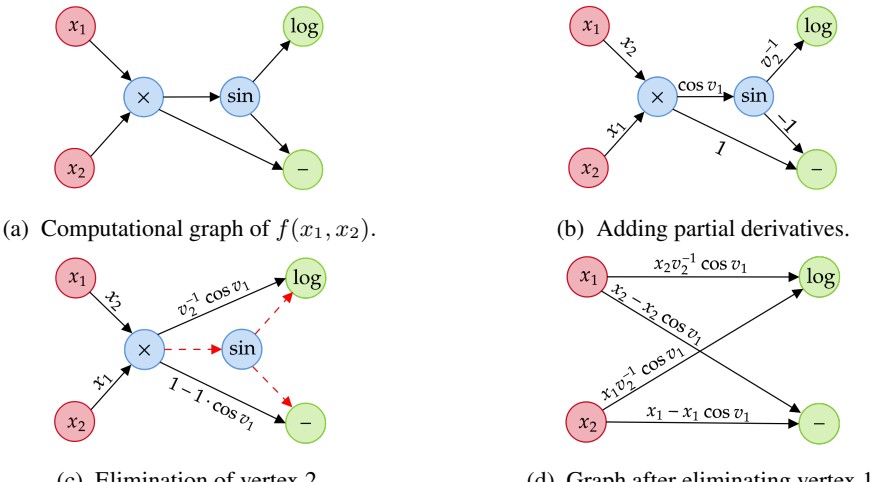

(a) Computational graph of $f(x_1, x_2)$.

(b) Adding partial derivatives.

(c) Elimination of vertex 2.

(d) Graph after eliminating vertex 1.

Figure 2: Step-by-step description of cross-country elimination with the simple example function $f(x_1, x_2) = (\log \sin(x_1 x_2), x_1 x_2 - \sin(x_1 x_2))^\top$. (a) Initial computational graph. (b) The partial derivatives are added to the edges of the computational graph. The intermediate variables $v_1$ and $v_2$ are defined through $v_1 = x_1 x_2$ and $v_2 = \sin v_1$. (c) Elimination of vertex 2 associated with the sin operation. The dotted red lines represent the edges that are deleted. (d) Final bipartite graph after both intermediate vertices have been eliminated. All remaining edges contain entries of the Jacobian.

many numerical scientific problems and machine learning [Baydin et al., 2018, Griewank and Walther, 2008]. AD leverages the fact that most computer programs can be broken down into a sequence of simple elemental operations, for example additions, multiplications and trigonometric functions. Partial derivatives of these elemental operations are coded into the AD software and the Jacobian is accumulated by recursively applying the chain rule to the evaluation procedure. Since the partial derivatives are known up to machine precision, AD gives the Jacobian up to machine precision.

## 2.1 Graph View and Vertex Elimination

We take the graph view of AD where a function is defined through its computational graph $\mathcal{G} = (V, E)$ with its vertices $\mathcal{V}$ being the elemental operations $\phi_j$ and directed edges $\mathcal{E}$ that describe the data dependencies between the operations (figure 2a). The relation $i \prec j$ states that vertex $i$ has an edge connecting it with vertex $j$, meaning that the output $v_i$ of $\phi_i$ is an input of $\phi_j$. The partial derivatives of the elemental operations with respect to their dependents are assigned to the connecting edges (figure 2b). We can then identify the edges of the graph with their respective partial derivatives $c_{ji} = \frac{\partial \phi_j}{\partial v_i}$. The cross-country elimination algorithm computes the Jacobian by a procedure called *vertex elimination*.

**Definition 1** [Griewank and Walther, 2008] *For a computational graph $\mathcal{G} = (V, E)$ with partial derivatives $c_{ij}$, vertex elimination of vertex $j$ is defined as the update*

$$c_{ki} \mathrel{+}= c_{kj} c_{ji} \ \forall (i, k) \in \mathcal{V} \times \mathcal{V} \text{ where } i \prec j \text{ and } j \prec k \tag{1}$$

*and then setting $c_{ji} = c_{kj} = 0$ for all involved vertices. The $\mathrel{+}=$ operator creates a new edge if there is no edge $c_{ki}$ and otherwise adds the new value to the existing value.*

Intuitively, vertex elimination can be understood as the *local* application of the chain rule to a single vertex in the graph since the multiplication $c_{ij} c_{jk}$ is exactly the result of applying the chain rule to $(\phi_k \circ \phi_j)(v_i)$. If a vertex has multiple incoming and outgoing edges, all combinations of incoming and outgoing edges are resolved to create new edges. If an edge already exists, we add the result of the product to it, in accordance with the rules for total derivatives. After the new edges are added to the graph, we delete all edges connected to the eliminated vertex since all the derivative information is now contained in the new edges (figure 2c). Note that the new graph resulting from a vertex elimination no longer directly represents the data dependencies of the function since the eliminated vertex is now disconnected. Furthermore, the application of vertex elimination as described above

requires the computational graph to be static and precludes the use of control flow (*if*-statements, *for*-loops) in the differentiated code.

## 2.2 Cross Country and Elimination Orders

The repeated application of the vertex elimination procedure to a computational graph (*i.e.* cross country elimination) until all intermediate vertices are eliminated will yield a graph where the input vertices and output vertices are directly connected by edges (no intermediate vertices left, see figure 2d). This is called a *bipartite graph* and the edges of this graph contain the components of the Jacobian of the function $f$. In particular, as long as all intermediate vertices are eliminated, this Jacobian will always be exact up to machine precision [Griewank and Walther, 2008]. There is no restriction on the order in which the vertices are eliminated, but the choice will significantly influence computational cost and memory [Tadjouddine et al., 2006]. In the graph view, computational cost is straightforward to measure since every vertex elimination incurs a known number of multiplications that depends on the shapes of the elemental Jacobians which can be used as a proxy for execution time. We can ignore the cost of evaluating the partial derivatives since they have to be performed regardless of the elimination order. Thus, we use the number of multiplications as the optimization target for the remainder of this work. The two most common choices for elimination orders are to either eliminate the vertices in the forward or reverse order. These two modes are called forward-mode AD and reverse-mode AD (backpropagation), respectively.

Forward-mode AD, where vertices are eliminated in the same order as the computational graph is traversed, is particularly efficient for functions where the number of input variables $n$ is much smaller than the number of output variables $m$, i.e. $n \ll m$. In contrast, reverse-mode AD traverses the graph in the opposite direction and is particularly suited for the cases where $n \gg m$. This is the case in machine learning and neural networks using scalar loss functions, which is why reverse-mode AD is the default choice in such workloads.

## 2.3 Minimal Markowitz Degree

A more advanced technique is to eliminate vertices with the lowest Markowitz degree first Griewank and Walther [2008]. The Markowitz degree of a vertex is defined as the number of incoming vertices times the number of outgoing vertices, *i.e.* $\mathrm{Mark}(j) = |i \prec j||j \prec k|$, where $|\cdot|$ denotes the cardinality of the sets $i \prec j$ and $j \prec k$ for fixed $j$. Thus the elimination order is constrained by finding the vertex with the lowest Markowitz degree first, eliminating it and then finding the next vertex with minimal Markowitz degree on the resulting graph. This elimination scheme is one of the best known heuristics for finding efficient elimination orders and can incur savings of up to 20% over forward- and reverse-mode AD [Albrecht et al., 2003, Griewank and Walther, 2008]. However, for computational graphs that have many inputs and few outputs, it is often outperformed by reverse-mode AD.

## 2.4 Vector-valued Functions

In most applications, vector-valued functions are used as elemental building blocks of more complex functions. While in most cases, these vectorized operations could be broken down into scalar operations, this would be impractical since it would increase the size of the computational graph representation and action space by orders of magnitude. Thus, it is best to allow vertices of the computational graph to be vector-valued which results in the partial derivatives assigned to the edges becoming Jacobians in their own right. The multiplication operations during vertex elimination are then accordingly replaced with matrix multiplications or higher-order contractions of the elemental Jacobians. For many operations, the Jacobians themselves have a particular internal sparsity structure which can be exploited when performing the eliminations. A simple example is the multiplication of a vector with a matrix followed by the application of a non-linear function $f(\mathbf{x}, \mathbf{W}) = \tanh(\mathbf{W} \cdot \mathbf{x})$. The input vertices are given by the input $\mathbf{x}$ and weights $\mathbf{W}$ and the intermediate vertex is matrix multiplication $a_i = \sum_j W_{ij} x_j$ with the partial derivatives

$$\frac{\partial a_i}{\partial W_{kl}} = x_l \delta_{ik}, \qquad\qquad \frac{\partial a_i}{\partial x_k} = W_{ik}. \qquad (2)$$

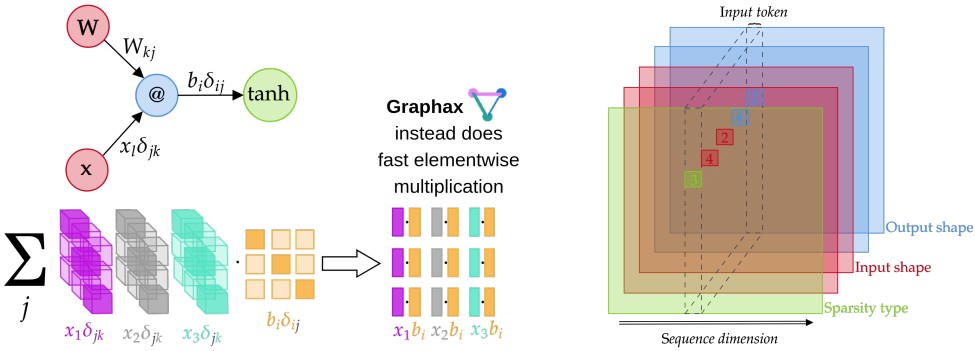

(a) Sparse vertex elimination.           (b) Computational graph representation.

Figure 3: (a) Graphax implements sparse vertex elimination to benefit from the advantages of cross country elimination. (b) Sketch of the three-dimensional adjacency tensor that represents the computational graph. The colored surfaces represent the five different values encoded in the third dimension. The red and blue surfaces together contain the shape of the Jacobians while the green surface encodes their sparsity. The vertical dotted slices represent the input connectivity of a single vertex. In this work, we compress and feed the vertical slices as tokens into the transformer backbone such that we build a sequence running in direction of the black arrow.

The output vertex represents the application of the activation function $y_i = \tanh a_i$ with the partial derivative

$$\frac{\partial y_i}{\partial a_j} = \delta_{ij}(1 - \tanh^2 a_i). \tag{3}$$

According to the vertex elimination rule, upon elimination of the intermediate vertex the two Jacobians in equation (2) are assigned to the incoming edges are contracted together with the Jacobian from the outgoing edge in equation (3):

$$\frac{\partial y_i}{\partial W_{kl}} = \sum_j (1 - \tanh^2 a_i)\delta_{ij}\delta_{jk}x_l = \delta_{ik}(1 - \tanh^2 a_i)x_l, \tag{4}$$

$$\frac{\partial y_i}{\partial x_k} = \sum_j (1 - \tanh^2 a_i)\delta_{ij}W_{jk} = (1 - \tanh^2 a_i)W_{ki}. \tag{5}$$

In both cases, instead of a matrix multiplication, one can perform simple element-wise multiplications as shown in figure 3a. For vectorized cross-country elimination to be efficient, it is paramount to exploit this property. Current state-of-the-art AD frameworks typically lack the ability to perform cross-country elimination and subsequently can not deal with sparse Jacobians. The only exception the authors are aware of is EliAD, an AD interpreter in C++ which is fully capable of processing given elimination orders and create the derivative source code [Tadjouddine et al., 2002a]. However, we developed *Graphax* as a novel AD interpreter that builds on Google's JAX [Bradbury et al., 2018] in order to leverage it's defining features such as JIT compilation, automated batching, device parallelism and a user-friendly Python front-end. Graphax is a fully fledged AD interpreter capable of performing cross-country elimination as described above and outperforms JAX' AD on the relevant tasks by several orders of magnitude (appendix B). Graphax and AlphaGrad are available under and https://github.com/jamielohoff/graphax and https://github.com/jamielohoff/alphagrad.

## 2.5 Computational Graph Representation and Network Architecture

We describe here how the computational graph is represented for optimization in the RL algorithm, as well as the network architecture that is optimized with AlphaZero. In the scalar case, the computational graph can be represented by its adjacency matrix, meaning that for every pair of vertices $(i, j)$ that share an edge, we set the $i$-th row and the $j$-th column of the matrix to 1. For the vectorized case, we define an extended adjacency tensor by extending the matrix into the third dimension. Along this third dimension, we store 5 values that describe the sparsity pattern and shape of the Jacobian associated with the respective edge. The first value is an integer between -10 and 10 which encodes the sparsity type of the Jacobian. Details about the supported sparsity types can

be found in Appendix C. The next four values contain the shape of the Jacobian associated with the respective edge and thus imply that this representation can at most deal with Jacobians of the shape $\frac{\partial y_{ij}}{\partial x_{kl}}$ where the first two values describe the shape of $x_{kl}$ and the other two values describe the shape of $y_{ij}$. This can be expanded to arbitrary tensor sizes of $x$ and $y$, but then also requires the definition of new sparsity types to account for the additional dimensions. Figure 3b shows the representation of the entire computational graph and a single selected edge with a Jacobian of shape $(4, 2, 4, 2)$ with sparsity type 3. A horizontal or vertical slice of the extended adjacency tensor gives the input or output connectivity of a particular vertex. These slices can be compressed and used as tokens to be fed into a transformer where they are processed simultaneously by the attention mechanism so that the model gets a full view of the graph's connectivity.

In this work, we compress vertical slices into tokens using a convolutional layer with kernel size (3, 5) and use a linear projection to create a 64-dimensional embedding. We found it helpful to apply a positional encoding to the tokens [Vaswani et al., 2017]. The output of the transformer is then fed into a policy and a value head. The policy head is a MLP mapped across every token separately, thus creating a probability distribution over the vertices to determine the next one to be eliminated. Already eliminated vertices are masked. Similarly, the value head is also a MLP that predicts a score for every token. These scores are then summed to give the value prediction of the network.

## 3 Reinforcement Learning for Optimal Elimination Orders

Cross-country elimination is typically introduced as a means to reduce the computational cost of computing the Jacobian. We cast the problem of finding an efficient vertex elimination order as a single-player RL game called *VertexGame*. At every step of the game, the agent selects the next vertex to be eliminated by observing the current connectivity of the computational graph. Since it is difficult to directly optimize for execution time, it is common to use the number of multiplications incurred by the elimination order as a proxy value [Tadjouddine et al., 2006, 2002b, Albrecht et al., 2003]. Thus, we chose the negative number of multiplications incurred by eliminating the selected vertex as reward. We use action masking to prevent the agent from eliminating the same vertex twice. This also ensures that the accumulated Jacobian is always exact and has a clear terminal condition: when the extended computational graph is bipartite, the game ends.

Between elimination orders, the magnitude of the reward can range across multiple orders of magnitude. To tackle this, we rescale the cumulative reward using a monotonous function. For the functions with scalar inputs as well as *RoeFlux_3d* and *random function* $f$, we found the method presented in [Kapturowski et al., 2019] performed well, *i.e.* we scaled with $s(r) = \text{sgn}(r)(\sqrt{|r| + 1} - 1) + \epsilon r$ where $\epsilon = 10^{-3}$. For *MLP* and *TransformerEncoder* tasks, the best performance was achieved with logarithmic scaling $s(r) = \log r$ [Hafner et al., 2024]. VertexGame is played by an AlphaZero agent, which successfully finds new AD algorithms. To reduce the computational cost of the AlphaZero agent [Silver et al., 2017], we employed Gumbel action sampling Danihelka et al. [2022]. Gumbel AlphaZero is a policy improvement algorithm based on sampling actions without replacement which utilizes the Gumbel softmax trick and other augmentations. This algorithm is guaranteed to improve the policy while significantly reducing the number of necessary Monte-Carlo Tree Search (MCTS) simulations. On most tasks, we found that 50 MCTS simulations were sufficient to reach satisfactory performance. Appendix D contains more details about the training of the agent.

## 4 Experiments

To demonstrate the effectiveness of our approach, we devised a set of tasks sampled from different scientific domains where AD is used to compute Jacobians. More details concerning the tasks are listed in appendix A.

**Deep Learning** is a prime example for the success of large-scale AD. We analyze a *two-layer MLP* with layer norm as described in [Goodfellow et al., 2016] and a small-scale version of the *transformer encoder* [Dosovitskiy et al., 2020].

**Computational Fluid Dynamics** relies on AD for computation of the flux Jacobian on the boundaries of the simulation grid cells. The *RoeFlux* is particularly relevant and has been studied extensively with vertex elimination in the past [Roe, 1981, Tadjouddine et al., 2002b, Zubair et al., 2023]. We test on the 1D and the 3D variants of this problem.

**Differential Kinematics** uses Jacobians to quantify the behavior of a robot or other mechanical system with respect to their controllable parameters (e.g. joints, actuators). There has been a surge

Table 1: Number of multiplications required by the best discovered elimination order for a batch size of one. Results obtained from VertexGame played by the AlphaZero agent with 50 MCTS simulations and a Gumbel noise scale of 1.0. † marks the experiments where we employed a log-scaling of the cumulative reward instead of the default scaling. The values in parentheses were obtained for 250 MCTS simulations.

| Task | Forward | Reverse | Markowitz | AlphaGrad |
|---|---|---|---|---|
| RoeFlux_1d | 620 | 364 | 407 | 320 |
| RobotArm_6DOF | 397 | 301 | 288 | 231 |
| HumanHeartDipole | 240 | 172 | 194 | 149 |
| PropaneCombustion | 151 | 90 | 111 | 88 |
| Random function $g$ | 632 | 566 | 451 | 417 |
| BlackScholes | 545 | 572 | 350 | 312 |
| RoeFlux_3d | 1556 | 979 | 938 | 811 |
| Random function $f$ | 17728 | 9333 | 12083 | 6374 |
| 2-layer MLP† | 10930 | 392 | 4796 | 398 (389) |
| Transformer† | 135010 | 4688 | 51869 | 4831 (4656) |

in interest of computing the Jacobian using AD[Giftthaler et al., 2017]. We chose the forward kinematics of a *6-DOF robot arm* as a representative problem and follow [Dikmenli, 2022] for the implementation.

**Non-Linear Equation Solving** requires the computation of large Jacobians to apply state-of-the-art solvers. The MINPACK problem collection provides a set of problems derived from real-life applications of non-linear optimization and designed to be representative of commonly encountered problems. In particular, we analyze the *HumanHeartDiple* and *PropaneCombustion* tasks, for which vertex elimination has also been analyzed thoroughly in [Forth et al., 2004b, Averick et al., 1992].

**Computational Finance** makes use of AD for fast computation of the so called "greeks" which measure the sensitivities of the value of an option to the model parameters[Naumann, 2010, Savine and Andreasen, 2021]. Here, we compute the second-order greeks of the *Black-Scholes equation* using AD by computing the Hessian of the Black-Scholes equation through evaluation of the Jacobian of the Jacobian Black and Scholes [1973]. This way, this task serves a two-fold purpose by also demonstrating how our approach is also useful for finding good AD algorithms for higher-order derivatives.

**Random Functions** are also commonly used to evaluate the performance of new AD algorithms [Albrecht et al., 2003]. We generated two random functions $f$ and $g$ with vector-valued and only scalar inputs respectively. The random code generator used to generate these arbitrary functions is included in the accompanying software package.

## 4.1 Finding Optimal Elimination Orders

Table 1 shows the number of multiplications required by the best elimination order found over 6 runs with different seeds. The model was trained from scratch on each task separately with a batch size of 1 to to keep the rewards as small as possible. The resulting AD algorithms are nonetheless scalable to arbitrary batch sizes. We use forward-mode, reverse-mode and the minimal Markowitz degree method as baselines for comparison. The first six tasks are simple functions with only scalar inputs and simple operations and the cumulative reward stays within the same order of magnitude, making them easier to solve. For all tasks, our approach was able find new elimination orders with improvements ranging from 2% to almost 20%. We found that even for only 5 MCTS simulations, the agent was able to find better than state-of-the-art solutions for the scalar tasks.

On the opposite spectrum, our experiments with 250 MCTS simulations yielded no significant improvement over the results presented in table 1. The four remaining tasks are arguably more difficult since the vector-valued inputs and large variance within possible rewards provide an additional challenge. The *RoeFlux_3d* and *random function f* were solved successfully with 50 MCTS simulations and yielded improvements of up to 33%. This is in stark contrast to prior work such as [Naumann, 1999], where algorithms such as simulated annealing or dynamic programming struggled to even beat common heuristics such as minimal Markowitz or reverse-mode AD. With a budget of only 50 MCTS simulations, AlphaGrad failed to find improvements for both deep learning tasks.

Table 2: Median runtimes for the results obtained in table 1. Results were measured with Graphax for batch size 512 on an AMD EPYC 9684X 2x96-Core processor. Uncertainties are given as 2.5- and 97.5-percentiles over 1000 trials. Execution time is given in milliseconds and default XLA compilation flags were used for all experiments. The size of the networks, i.e. the number of neurons were increased for the MLP and Transformer Encoder by a factor of 16 to create a more realistic sample. GPU experiments were run on a NVIDIA RTX 4090 with JIT compilation.

| Task | Forward | Reverse | Markowitz | AlphaGrad |
|---|---|---|---|---|
| RoeFlux_1d | $3.03^{+0.17}_{-0.27}$ | $3.08^{+0.17}_{-0.23}$ | $2.87^{+0.22}_{-0.66}$ | $2.19^{+0.30}_{-0.35}$ |
| RobotArm_6DOF | $8.85^{+0.21}_{-0.17}$ | $8.48^{+0.32}_{-0.20}$ | $8.55^{+0.38}_{-0.36}$ | $6.05^{+0.34}_{-0.35}$ |
| HumanHeartDipole | $16.97^{+1.23}_{-2.39}$ | $16.87^{+1.45}_{-3.90}$ | $16.42^{+1.29}_{-2.73}$ | $15.94^{+1.11}_{-1.10}$ |
| PropaneCombustion | $36.91^{+2.87}_{-1.50}$ | $36.47^{+1.45}_{-0.51}$ | $36.99^{+1.55}_{-1.11}$ | $36.45^{+2.47}_{-1.31}$ |
| Random function $g$ | $82.41^{+2.85}_{-2.97}$ | $81.64^{+2.82}_{-3.56}$ | $82.97^{+1.38}_{-1.17}$ | $80.56^{+1.61}_{-2.30}$ |
| BlackScholes | $5.04^{+0.23}_{-0.33}$ | $5.02^{+0.30}_{-0.39}$ | $5.03^{+0.23}_{-0.29}$ | $4.77^{+0.23}_{-0.29}$ |
| RoeFlux_3d | $72.26^{+3.22}_{-6.02}$ | $83.98^{+5.69}_{-6.68}$ | $91.27^{+11.59}_{-16.49}$ | $63.92^{+4.45}_{-5.76}$ |
| Random function $f$ | $12.42^{+0.66}_{-0.25}$ | $11.54^{+0.15}_{-0.27}$ | $20.20^{+0.26}_{-0.31}$ | $9.12^{+0.08}_{-0.06}$ |
| 2-layer MLP† | $760.63^{+80.01}_{-53.30}$ | $29.67^{+1.33}_{-4.05}$ | $317.65^{+53.30}_{-19.34}$ | $28.73^{+1.32}_{-3.78}$ |
| 2-layer MLP†(GPU) | $11.04^{+0.31}_{-0.13}$ | $0.30^{+0.02}_{-0.03}$ | $1.01^{+0.02}_{-0.05}$ | $0.29^{+0.02}_{-0.03}$ |
| Transformer† | $990.09^{+35.42}_{-31.11}$ | $39.07^{+4.59}_{-7.67}$ | $498.94^{+20.34}_{-19.62}$ | $40.38^{+5.26}_{-3.62}$ |
| Transformer†(GPU) | $21.38^{+0.22}_{-0.06}$ | $0.29^{+0.02}_{-0.03}$ | $16.17^{+15.04}_{-0.09}$ | $0.28^{+0.02}_{-0.01}$ |

The authors conjecture that this is not only due to difficulties presented above, but because reverse-mode AD (backpropagation) is already a very well-suited algorithm for computing Jacobians of "funnel-like" computational graphs with many inputs and a single, scalar output. Despite this, with an increase to 250 MCTS simulations, the agent marginally outperformed backpropagation for both deep learning models. Appendix E contains more information about the experiments, including reward curves, the actual elimination orders and more details about their implementation. Note that the results in table 1 were obtained by separately training on each single function/graph.

We also include joint training runs where the agent was trained on all tasks at once. While the results were inferior to the separate training mode, the agent found new, improved elimination orders for almost all tasks except the *MLP*, *TransformerEncoder*, *RoeFlux_3d* and *PropaneCombustion* tasks. For the *random function f* and *BlackScholes_Jacobian* task, the multi-task training outperformed the results in table 1 with new best results of 5884 and 307 respectively, thereby showing that the algorithm search might benefit from training on diverse tasks simultaneously. This also hints at the possibility of building a more general statistical model of AD applicable to workloads from many different domains. We also experimented with PPO as an alternative (appendix F).

## 4.2 Runtime Improvements and the Graphax library

The results in table 1 are mainly of theoretical value. Here, we investigate how these translate into actual runtime improvements. For this purpose, we implemented Graphax, to our knowledge the first Python-based AD interpreter able to leverage cross-country elimination. Graphax builds a second program that computes the Jacobian by leveraging the elimination orders found by AlphaGrad and using the source code of the function as a template by analyzing its *Jaxpression*. The Jaxpression is JAX' own representation of the computational graph of the function in question.

Table 2 shows runtime improvements for the elimination orders found in table 1 for a batch size of 512 with varying levels of improvement. This is due to the fact that the number of multiplications alone was only a proxy to capture the complexity of the entire program. It ignores other relevant quantities such as memory accesses and operation fusion during compilation. Nonetheless, a particularly impressive gain over the state-of-the-art methods can be observed for the *RoeFlux* tasks and the *RobotArm_6DOF* task. Remarkably, we also observe a minor improvement for both deep learning tasks when executed on GPUs. Note that the *TransformerEncoder* task was evaluated on a batch size of 1 because VertexGame only supports two-dimensional input tensors. Figure 4 shows how some of AlphaGrad's algorithms scale with growing batch size. Appendix B provides an in-depth comparison of our work and JAX' own AD modes. In general, the combination of AlphaGrad and Graphax is able to outperform the JAX AD modes in most cases, sometimes by orders of magnitude. While

[Forth et al., 2004b] and [Tadjouddine et al., 2006] present state-of-the-art results for some of the investigated tasks, we were unable to reproduce the experiments given their implementation details.

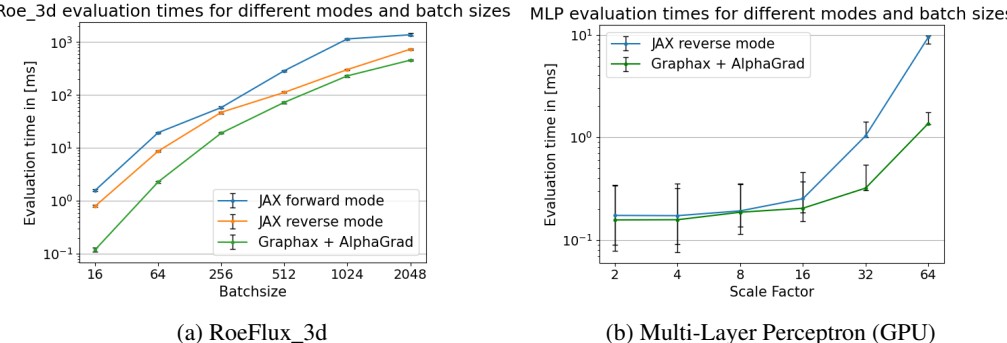

(a) RoeFlux_3d                (b) Multi-Layer Perceptron (GPU)

Figure 4: Runtime measurements over 1000 trials for the vectorized *RoeFlux_3d* and *MLP* tasks with different batch sizes using the same setup as in table 2. The MLP network sizes were scaled up with growing batch size by a constant factor. The exact procedure of scaling is explained in appendix B. Error bars are the 2.5- and 97.5-percentiles of the runtimes.

## 5 Conclusion

In this work we successfully demonstrated that AlphaGrad discovers new AD algorithms that outperform the state-of-the-art. We demonstrated that these theoretical gains translate into measurable runtime improvements with Graphax, a Python-based interpreter we developed that leverages the AD algorithms discovered by AlphaGrad. However, AlphaGrad currently only optimizes for multiplications which cannot capture the entire complexity of the AD algorithm. Future work could explore other optimization targets such as execution time, memory accesses, quantization and different hardware backends using a hardware model for efficient simulation. Another promising research avenue would be the implementation of a much more general framework for vertex elimination. Inspiration could be drawn from Enzyme, which operates on the intermediate representations (IR) using LLVM and is therefore less bound by the choice of programming language. Leveraging the LLVM approach would also open up new directions regarding AD-specific compiler optimizations similar to what was presented in LAGrad.

Two of the main shortcomings of our work are the lack of support for dynamic control flow and dynamically changing functions as well as the need for retraining of the algorithm for every single computational graph. To circumvent the first issue, it would be possible to implement a version of vertex elimination that can deal with dynamic control flow although this would require some significant changes to Graphax. However, as demonstrated by the wide range of benchmark tasks, several applications are already possible without this feature.

The second issue, addressed partially in our work, is the training of the agent on multiple graphs at once. While the best results were still achieved with single-graph training, the multi-graph training still outperformed the other existing methods on most benchmarks. This hints at the possibility to train our agent on a large set of computational graphs at once, thereby effectively building a statistical model of AD. Finally, VertexGame offers a novel way to evaluate existing RL algorithms on a real-world problem that poses diverse challenges such as rewards across multiple scales and large action spaces. Thus, it could complement existing benchmarks such as OpenAI gym and MuJoCo [Todorov et al., 2012, Towers et al., 2023].

## Acknowledgments

This work was sponsored by the Federal Ministry of Education, Germany (projects NEUROTEC-II grant no. 16ME0398K and 16ME0399 as well as GreenEdge-FuE, funding no. 16ME0521). The authors also gratefully acknowledge the Gauss Centre for Supercomputing e.V. (www.gauss-centre.eu) for funding this project by providing computing time on the GCS Supercomputer JUWELS at Jülich Supercomputing Centre (JSC). Furthermore, we thank Mark Schöne, Christian Pehle, Uwe Naumann and Matthew Johnson for the helpful discussions regarding the project.

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

# Appendix

## A    Task Descriptions

This section describes in-depth the mathematical formulation of the tasks that were evaluated in table 1 and table 2. An implementation of the functions can be found in the accompanying software package. Note that if a set of variables $(x_1, \ldots, x_n)$ is referred to as $\{x_i\}$, they are treated as a separate scalar inputs to the function while x treats them as a single vectorized input.

### A.1    RoeFlux_1d

For the implementation of the *RoeFlux_1d* task we thoroughly followed [Roe, 1981]. The pressure $p_{1\mathrm{d}}$, enthalpy $H_{1\mathrm{d}}$ and flux term $F_{1\mathrm{d}}$ of the one-dimensional Euler equations are defined through:

$$p_{1\mathrm{d}}(u_0, u_1, u_2) = (\gamma - 1)(u_2 - \frac{u_1^2}{2u_0}),$$

$$H_{1\mathrm{d}}(u_0, u_2, p) = \frac{u_2 + p}{u_0},$$

$$F_{1\mathrm{d}}(u_0, u_1, u_2, p) = \left(u_1, p + \frac{u_1^2}{u_0}, \frac{u_1}{u_0}(p + u_2)\right).$$

The Roe flux $\phi_{\mathrm{Roe}}(u_{\mathrm{l}0}, u_{\mathrm{l}1}, u_{\mathrm{l}2}, u_{\mathrm{l}3}, u_{\mathrm{l}4}, u_{\mathrm{r}0}, u_{\mathrm{r}1}, u_{\mathrm{r}2}, u_{\mathrm{r}3}, u_{\mathrm{r}4})$ between two adjacent cells $u_{\mathrm{l}}$ and $u_{\mathrm{r}}$ is computed using the routine described below. First, we define some averaged quantities to simplify formulation:

$$\Delta u_0 = u_{\mathrm{l}0} - u_{\mathrm{r}0},$$
$$u_{\mathrm{lr}0} = \sqrt{u_{\mathrm{l}0} u_{\mathrm{r}0}},$$
$$w_1 = \sqrt{u_{\mathrm{l}0}} - \sqrt{u_{\mathrm{r}0}}.$$

Then we define some state variables for the left cell through

$$v_{\mathrm{l}} = \frac{u_{\mathrm{l}1}}{u_{\mathrm{l}0}},$$
$$p_{\mathrm{l}} = p_{1\mathrm{d}}(u_{\mathrm{l}0}, u_{\mathrm{l}1}, u_{\mathrm{l}2}),$$
$$h_{\mathrm{l}} = H_{1\mathrm{d}}(u_{\mathrm{l}0}, u_{\mathrm{l}2}, p_{\mathrm{l}}).$$

Similarly, we define the same variables for the right cell through

$$v_{\mathrm{r}} = \frac{u_{\mathrm{r}1}}{u_{\mathrm{r}0}},$$
$$p_{\mathrm{r}} = p_{1\mathrm{d}}(u_{\mathrm{r}0}, u_{\mathrm{r}1}, u_{\mathrm{r}2}),$$
$$h_{\mathrm{r}} = H_{1\mathrm{d}}(u_{\mathrm{r}0}, u_{\mathrm{r}2}, p_{\mathrm{r}}).$$

We define some differences between the state variables of the two cells though

$$\Delta p = p_{\mathrm{l}} - p_{\mathrm{r}}, \quad \Delta v = v_{\mathrm{l}} - v_{\mathrm{r}},$$

We proceed with the introduction of some auxiliary variables for further computation so that

$$u = \frac{\sqrt{u_{\mathrm{l}0}} v_{\mathrm{l}} + \sqrt{u_{\mathrm{r}0}} v_{\mathrm{r}}}{w_1},$$
$$h = \frac{\sqrt{u_{\mathrm{l}0}} h_{\mathrm{l}} + \sqrt{u_{\mathrm{r}0}} h_{\mathrm{r}}}{w_1},$$
$$a_2 = (\gamma - 1)(h - \frac{1}{2}q_2),$$

where $q_2 = u^2$, $a = \sqrt{a_2}$ and $n = u_{\mathrm{lr}0} a$. We define $l_{\mathrm{p}} = |u + a|$, $l = |u|$ and $l_{\mathrm{n}} = |u|$ and proceed to writing:

$$c_0 = \left(\Delta u_0 - \frac{\Delta p}{a_2}\right) l$$

$$c_1 = \left(\Delta v + \frac{\Delta p}{n}\right) l_{\mathrm{p}}$$

$$c_2 = \left(\Delta v - \frac{\Delta p}{n}\right) l_{\mathrm{n}}$$

Next, we compute the fluxes between the cells through

$$F_{l0}, F_{l1}, F_{l2} = F_{1d}(u_{l0}, u_{l1}, u_{l2}, p_l)$$
$$F_{r0}, F_{r1}, F_{r2} = F_{1d}(u_{r0}, u_{r1}, u_{r2}, p_r)$$

and then define $F_i = F_{li} + F_{ri}$ and $\alpha = \frac{1}{2a}u_{lr0}$. The flux differences are then given through

$$\Delta F_0 = c_0 + \alpha c_1 - \alpha c_2,$$
$$\Delta F_1 = c_0 u + \alpha c_1 (u + a) - \alpha c_2 (u - a),$$
$$\Delta F_2 = \frac{1}{2}c_0 q_2 + \alpha c_1 (h + ua) - \alpha c_2 (h - ua).$$

Then the output of the function is given through $\phi_0, \phi_1, \phi_2$ with $\phi_i = \frac{1}{2}(F_i - \Delta F_i)$.

## A.2 RoeFlux_3d

The implementation of the *RoeFlux_3d* task is similar to the *RoeFlux_1d* task. Again, we follow [Roe, 1981] for the implementation and start by defining functions for the pressure $p_{3d}$ and enthalpy $H_{3d}$:

$$p_{3d}(u_0, \mathbf{u}, u_4) = (\gamma - 1)\left(u_4 - \frac{|\mathbf{u}|^2}{2u_0}\right),$$
$$H_{3d}(u_0, u_4, p) = \frac{u_4 + p}{u_0}.$$

Note that now instead of a one-dimensional state variable $u_1$, we have a three-dimensional state vector $\mathbf{u}$ and the role of $u_2$ is now taken over by $u_4$. The flux in three dimensions is given through

$$F_{3d}(u_0, \mathbf{u}, u_4, \mathbf{v}, p) = (u_1, \mathbf{p} + \mathbf{u}v, v_1(p + u_4))$$

where $u_1, u_2, u_3$ and $v_1, v_2, v_3$ are the three components that make up $\mathbf{u}$ and $\mathbf{v}$ respectively and $\mathbf{p} = (p, 0, 0)^\intercal$. We then again define the finite differences

$$\Delta u_0 = u_{l0} - u_{r0},$$
$$\Delta \mathbf{u} = \mathbf{u}_l - \mathbf{u}_r,$$
$$\Delta u_4 = u_{l4} - u_{r4}$$

and furthermore set $\mathbf{v}_l = \frac{\mathbf{u}_l}{u_{l0}}$ and $\mathbf{v}_r = \frac{\mathbf{u}_r}{u_{r0}}$ We continue with defining some auxiliary variables

$$w_1 = \sqrt{u_{l0}} - \sqrt{u_{r0}}.$$
$$\mathbf{t} = \frac{\sqrt{u_{l0}}\mathbf{v}_l + \sqrt{u_{r0}}\mathbf{v}_l}{w_1}.$$

Then we define some state variables for the left cell through

$$p_l = p_{3d}(u_{l0}, \mathbf{u}_l, u_{l4}),$$
$$h_l = H_{3d}(u_{l0}, u_{l4}, p_l).$$

Similarly, we define the same variables for the right cell through

$$p_r = p_{3d}(u_{r0}, \mathbf{u}_r, u_{r2}),$$
$$h_r = H_{3d}(u_{r0}, u_{r4}, p_r).$$

Then we introduce

$$h = \frac{\sqrt{u_{l0}}h_l + \sqrt{u_{r0}}h_r}{w_1},$$

and set $q_2 = |\mathbf{t}|^2$ and $a_2 = (\gamma - 1)(h - \frac{1}{2}q_2)$ and $a = \sqrt{a_2}$. Furthermore, we define the eigenvalues of the Roe flux problem as $l_{\mathrm{p}} = t_1 + a$, $l = t_1$, $l_{\mathrm{n}} = t_1 - a$. Then coefficients $c_i$ are given through

$$c_0 = \frac{k_1 - k_2}{2} l_{\mathrm{m}},$$

$$c_1 = l\left(\frac{\Delta u_2}{t_2} - \Delta u_0\right),$$

$$c_2 = l\left(\frac{\Delta u_3}{t_3} - \Delta u_1\right),$$

$$c_3 = l\left(\frac{\gamma - 1}{a_2}\left((h - q_2)\Delta u_0 + \mathbf{t} \cdot \Delta\mathbf{u} - \Delta u_4\right)\right),$$

$$c_4 = \frac{k_1 + k_2}{2} l_{\mathrm{p}}.$$

where we defined $k_1 = \Delta u_0 - c_3$ and $k_2 = \dfrac{\Delta u_1 - t_1 \Delta u_0}{a}$. The definition of the fluxes of each cell is similar to the formulation in the one-dimensional case. The flux changes are given through

$$\Delta F_0 = c_0 + c_3 + c_4 l_{\mathrm{p}},$$
$$\Delta F_1 = c_0 l_{\mathrm{n}} + c_3 t_1 + c_4 l_{\mathrm{p}},$$
$$\Delta F_2 = c_0 t_2 + c_1 t_2 + c_2 t_2 + c_3 t_2 + c_4 t_2,$$
$$\Delta F_3 = c_0 t_3 + c_2 t_3 + c_3 t_3 + c_4 t_3,$$
$$\Delta F_4 = c_0(h - t_1 a) + c_1 t_2^2 + c_2 t_3^2 + \frac{c_3 q_2}{2} + c_4(h + t_1 a).$$

The results of $\Delta F_1$, $\Delta F_2$ and $\Delta F_3$ are concatenated into the vector $\Delta\mathbf{F}$. The output of the function is then $(\phi_0, \phi, \phi_4)$ as defined in the one-dimensional case.

## A.3 RobotArm_6DOF

The *RobotArm_6DOF* task models the forward differential kinematics of a 6-degree-of-freedom (6-DOF) robot arm as is often found in robotics labs and industrial manufacturing sites. For the implementation, we followed [Dikmenli, 2022] and define $c_i = \cos x_i$ and $s_i = \sin x_i$. Furthermore, we define the functions

$$S(u, v) = \cos(u)\sin(v) + \sin(u)\cos(v),$$
$$C(u, v) = \cos(u)\cos(v) - \sin(u)\sin(v).$$

Then we also define $s_{ij} = S(t_i, t_j)$ and $c_{ij} = C(t_i, t_j)$ for the input variables $\{t_i\}$ of the problem. Then we proceed with calculating the auxiliary intermediates

$$a_y = s_5(c_1 c_{23} c_4 + s_1 s_4) + c_1 s_{23} c_5,$$
$$a_y = s_5(s_1 c_{23} c_4 - c_1 s_4) + s_1 s_{23} c_5,$$
$$a_z = s_{23} c_4 s_5 - c_{23} c_5,$$
$$n_z = c_6(c_{23} s_5 + s_{23} c_4 c_5) - s_{23} s_4 s_6,$$
$$o_z = -s_6(c_{23} s_5 + s_{23} c_4 c_5) - s_{23} s_4 c_6.$$

Then, we define the *Tait-Bryan angles* through:

$$z = \arctan \frac{a_y}{a_x}$$

$$\hat{y} = \arctan \frac{\sqrt{1 - a_z^2}}{a_z}$$

$$\hat{z} = \arctan\left(-\frac{o_z}{n_z}\right)$$

Next, we calculate the positional parts of the kinematics by starting with the $x$-component:

$$x_1 = 185(s_5(c_1 c_{23} c_4 + s_1 s_4) + c_1 s_{23} c_5),$$
$$x_2 = c_1(175 + 890 c_2 + 50 c_{23} + 1035 s_{23}),$$
$$p_x = x_1 + x_2$$

We continue with the $y$-component:

$$y_1 = 185(s_5(c_1c_{23}c_4 + c_1s_4) + s_1s_{23}c_5),$$
$$y_2 = s_1(175 + 890c_2 + 50c_{23} + 1035s_{23}),$$
$$p_y = y_1 + y_2$$

Finally, the $z$-component is given through:

$$p_z = 575 + 890s_2 + 50s_{23} - 1035c_{23} + 185(s_{23}c_4s_5 - c_{23}c_5)$$

The function then returns the six values $(p_x, p_y, p_z, \hat{z}, \hat{y}, \hat{z})$

## A.4 HumanHeartDipole

The *HumanHeartDipole* task is derived from the experimental electrolytic determination of the resultant dipole moment in the human heart. For the implementation, we followed [Averick et al., 1992] with $\{x_i\} = (x_1, \ldots, x_8)$ such that

$f_1(\{x_i\}) = x_1 + x_2 - \sigma_{\text{mx}}$

$f_2(\{x_i\}) = x_3 + x_4 - \sigma_{\text{my}}$

$f_3(\{x_i\}) = x_5x_1 + x_6x_2 - x_7x_3 - x_8x_4 - \sigma_{\text{A}}$

$f_4(\{x_i\}) = x_7x_1 + x_8x_2 + x_5x_3 + x_6x_4 - \sigma_{\text{B}}$

$f_5(\{x_i\}) = x_1(x_5^2 - x_7^2) - 2x_1x_5x_7 + x_2(x_6^2 - x_8^2) - 2x_4x_6x_8 - \sigma_{\text{C}}$

$f_6(\{x_i\}) = x_3(x_5^2 - x_7^2) + 2x_1x_5x_7 + x_4(x_6^2 - x_8^2) + 2x_2x_6x_8 - \sigma_{\text{D}}$

$f_7(\{x_i\}) = x_1x_5(x_5^2 - 3x_7^2) + x_3x_7(x_7^2 - 3x_5^2) + x_2x_6(x_6^2 - 3x_8^2) + x_4x_8(x_8^2 - 3x_6^2) - \sigma_{\text{E}}$

$f_8(\{x_i\}) = x_3x_5(x_5^2 - 3x_7^2) - x_1x_7(x_7^2 - 3x_5^2) + x_4x_6(x_6^2 - 3x_8^2) - x_2x_8(x_8^2 - 3x_6^2) - \sigma_{\text{F}}$

with $\sigma_{\text{mx}}, \sigma_{\text{my}}, \sigma_{\text{A}}, \sigma_{\text{B}}, \sigma_{\text{C}}, \sigma_{\text{D}}, \sigma_{\text{E}}, \sigma_{\text{F}}$ being some arbitrary measured constants.

## A.5 PropaneCombustion

The *PropaneCombustion* task arises in the determination of the chemical equilibrium of the combustion of propane in air. Each unknown is related to a product concentration given in mols formed during the combustion process. We implemented the *PropaneCombustion* task as defined in [Averick et al., 1992] where with $\{x_i\} := (x_1, \ldots, x_{11})$:

$$f_1(\{x_i\}) = x_1 + x_4 - 3$$
$$f_2(\{x_i\}) = 2x_1 + x_2 + x_4 + x_7 + x_8 + x_9 + 2x_{10} - R$$
$$f_3(\{x_i\}) = 2x_2 + 2x_5 + x_6 + x_7 - 8$$
$$f_4(\{x_i\}) = 2x_3 + x_9 - 4R$$
$$f_5(\{x_i\}) = K_5\sqrt{x_2x_4} + x_1x_5$$
$$f_6(\{x_i\}) = K_6\sqrt{x_1x_2} - \sqrt{x_4}x_7\sqrt{\frac{p}{x_{11}}}$$
$$f_7(\{x_i\}) = K_7\sqrt{x_1x_2} - \sqrt{x_4}x_7\sqrt{\frac{p}{x_{11}}}$$
$$f_8(\{x_i\}) = K_8x_1 - x_4x_8\frac{p}{x_{11}}$$
$$f_9(\{x_i\}) = K_9x_1\sqrt{x_3} - x_4x_9\sqrt{\frac{p}{x_{11}}}$$
$$f_{10}(\{x_i\}) = K_{10}x_1^2 - x_4^2x_{10}\frac{p}{x_{11}}$$
$$f_{11}(\{x_i\}) = x_{11} - x_{10} - x_9 - x_8 - x_7 - x_6 - x_5 - x_4 - x_3 - x_2 - x_1$$

Here, $K_5, \ldots, K_{10}$ are measured constants and $R = 10$ is the relative amount of air and fuel, while $p = 40$ is the pressure in atmospheres.

### A.6 Black-Scholes Equation

The *BlackScholes_Jacobian* task is derived from riskless portfolio management where the goal is the computation of so called second-order greeks which give insights about the price evolution of an option. The Black-Scholes partial differential equation is given through

$$\frac{\partial V}{\partial t} + \frac{1}{2}\sigma^2 S^2 \frac{\partial^2 V}{\partial S^2} = rV - rS\frac{\partial V}{\partial S}.$$

In this equation, $\sigma$ models the volatility of the underlying geometric Brownian motion, while $S$ describes the stock price of the underlying asset and $r$ is the risk-free interest rate. $V$ is the price of the option at time $t$. For certain conditions described in [Black and Scholes, 1973], this equation can be solved analytically. We define

$$\phi(x) = 1 + \frac{1}{\sqrt{\pi}} \int_{-\infty}^{x} e^{-\frac{1}{2}z^2} \mathrm{d}z \tag{6}$$

which is the cumulative distribution function of the Gaussian distribution (also known as the error function). Next, we define

$$d_1 = \frac{1}{\sigma\sqrt{T}} \left( \log\left(\frac{F}{K}\right) + \frac{\sigma^2}{2T} \right)$$
$$d_2 = d_1 - \sigma\sqrt{T}$$
$$\Phi(F, K, r, \sigma, T) = e^{-rT} \left( F\phi(d_1) - K\phi(d_2) \right)$$

where $K$ is the payoff. Then, the solution to the Black-Scholes equation is given through

$$f(S, K, r, \sigma, T) = \Phi(F(r, T, S), K, r, \sigma, T)$$

where $F(r, T, S) = e^{rT}S$. We are interested in the second-order derivatives with respect to $S, K, \sigma, r$, so we first calculate the Jacobian of equation (A.6) using reverse-mode AD with Graphax using *graphax.jacve(f, order="rev")*. The resulting computational graph is used to learn an optimal elimination order to compute the second-order derivatives with AD, i.e. the Hessian.

### A.7 Multi-Layer Perceptron

The *MLP* task describes a simple 2-layer Perceptron with a layer norm between the first and the second hidden layer. The input-size of the network is 4 and hidden layers have size 8 while the output layer has a size of 4. The activation functions are $\tanh$-functions and the output is processed using a softmax cross-entropy loss. For the actual runtime experiments, we scaled the sizes of the inputs, outputs and hidden layers by a factor of 16 to create a more realistic example of a MLP.

### A.8 Transformer Encoder

The *TransformerEncoder* task is inspired by recent advances in natural language processing and image classification. The network consists of two attention blocks with a single head only. The block itself consists of a softmax attention layer with residual connections followed by a layer norm and a MLP with a single hidden layer of size 4 and sigmoid linear unit activations. We stack two of these layers and process the output with a softmax cross-entropy loss function. The input embeddings have size 4 with a sequence length of 4. For the actual runtime experiments, we scaled the sizes of the inputs, outputs and hidden layers by a factor of 16 to create a realistic example of a transformer encoder.

### A.9 Random Functions f and g

The random functions $f$ and $g$ are randomly generated using a custom JAX interpreter which consists of a repository of elemental operations such as $\cos$, $\log$ or $+$ but also matrix multiplications and array reshape operations. The number of input and output variables can be specified as well as the number of intermediate vertices that will be eliminated by the vertex elimination algorithm. The random function generator then randomly samples elemental functions from the repository. The number of functions that are unary, binary or perform accumulation or reshape operations can be controlled by adjusting the respective sampling probabilities. At every step, the function is checked to guarantee that it is executable and well-defined. The function generator is part of the accompanying AlphaGrad software package.

# B   Comparison to JAX

Figures 5 and 6 contain an in-depth analysis of the performance benefits of the combination of AlphaGrad and Graphax over default JAX forward-mode AD and reverse-mode AD. Unless stated otherwise, all runtimes were measured on an AMD EPYC 9684X 2x96-Core processor. On the *RoeFlux_1d*, *RobotArm_6DOF*, *random function g* and *BlackScholes_Jacobian* tasks, our work significantly outperforms both JAX AD modes for all batch sizes, sometimes by almost an order of magnitude. For the *HumanHeartDipole* and *PropaneCombustion* tasks, AlphaGrad and Graphax outperform both JAX modes significantly but we can observe a crossover from batch size 1024 to 2048. For the *RoeFlux_3d* and *random function f* tasks, we found that the combination of AlphaGrad and Graphax consistently outperforms JAX' AD modes for large batch sizes.

The deep learning tasks, i.e. the *MLP* and *TransformerEncoder* tasks, were analyzed slightly differently. Both functions were vectorized only over their inputs and labels as is typically done in deep learning applications. Furthermore, we evaluated both networks for different scale factors where the entire network was scaled up by a constant factor and only compare to JAX reverse-mode AD since this is the default training mode for these kinds of networks. Also, both networks were evaluated on GPU as well as this is the typical hardware backend on which they are executed. For the *MLP* task we found that the AlphaGrad and Graphax combination outperforms JAX reverse-mode AD for large scale factors, i.e. large networks with a large batch size. Since the gain through AlphaGrad is small, the authors conjecture that the gain in performance is mainly due to the sparse implementation of the AD routines. Note that the batch size was also consistently scaled up with the other components of the network with a starting batch size of 8.

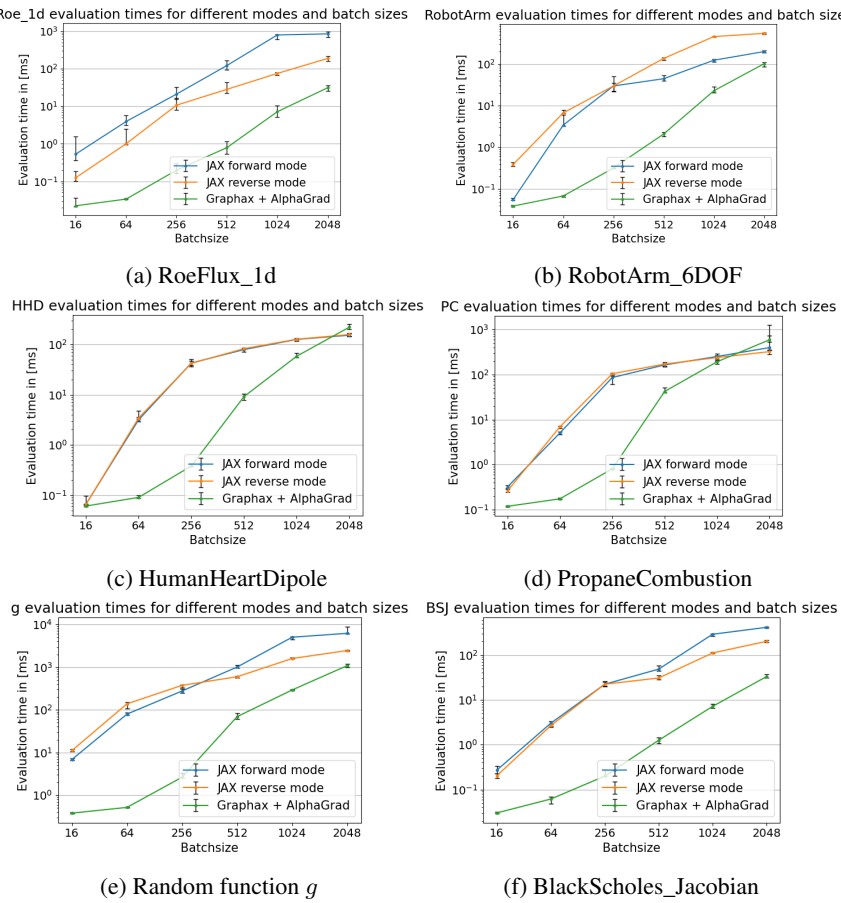

(a) RoeFlux_1d  (b) RobotArm_6DOF

(c) HumanHeartDipole  (d) PropaneCombustion

(e) Random function $g$  (f) BlackScholes_Jacobian

Figure 5: Runtime measurements over 100 trials for the scalar tasks. Error bars are the 2.5- and 97.5-percentiles of the runtimes.

For the *TransformerEncoder* task, we were only able to evaluate the runtime for a batch size of 1 since the VertexGame implementation of AlphaGrad supports only inputs with a maximum dimensionality of 2 while a typical batched transformer input has the shape (batch size, sequence length, embedding dimension). We found that the combination of AlphaGrad and Graphax performed on par with JAX' reverse-mode AD for all scale factors except 8 where we found a significant difference in performance.

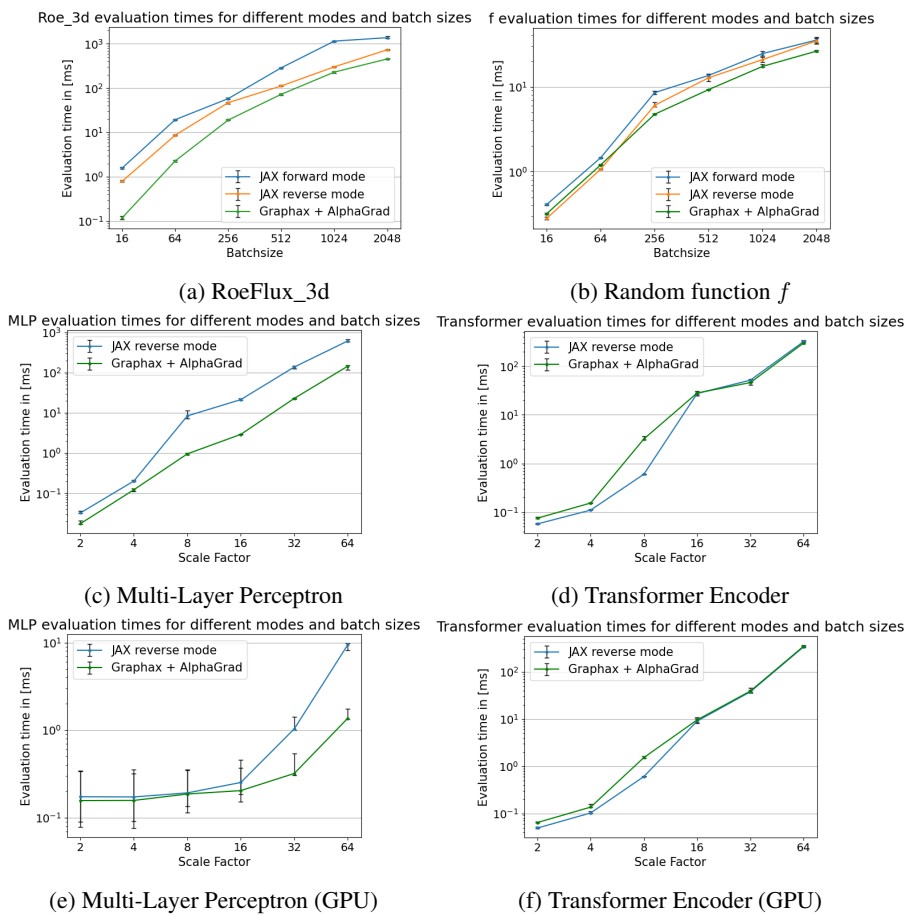

(a) RoeFlux_3d

(b) Random function $f$

(c) Multi-Layer Perceptron

(d) Transformer Encoder

(e) Multi-Layer Perceptron (GPU)

(f) Transformer Encoder (GPU)

Figure 6: Runtime measurements over 100 trials for the vectorized tasks with different batch sizes. The MLP network sizes were scaled up with growing batch size. The transformer could only be evaluated on with a batch size of 1 due to limitations of VertexGame but the network size was scaled up as well. Both deep learning tasks were also evaluated on GPUs. Error bars are the 2.5- and 97.5-percentiles of the runtimes.

## C  Sparsity Types

The sparsity of the Jacobians associated with the edges in the computational graph representation is described with a number ranging from -10 to 10. Table 3 contains an overview of the sparsity types and an example tensor that is represented by the corresponding number. Our approach is directed only at diagonal sparsity, meaning that we mainly consider tensors that can be decomposed into products of lower-dimensional tensors and Kronecker symbols $\delta_{ij}$. Since we might have operations like transposing or slicing in our computational graph, which just change the shape but not the value of the edge Jacobians when eliminated, we introduce a "copy gradient" sparsity type with value -1 to represent these operations. Furthermore, we make a distinction between multiplication with the unit tensor $\delta_{il}\delta_{jk}$ and a constant multiple of the unit tensor $c\delta_{il}\delta_{jk}$ since the latter incurs actual multiplications while the former can just be treated as a renaming of indices as demonstrated by the

Table 3: Different sparsity types that are required to represent all the possible tensor shapes that occur in tensor vertex elimination with vectors and matrices.

| Sparsity Type | Example Tensor |
| --- | --- |
| -10 | $c\delta_{il}\delta_{jk}$ |
| -9 | $T_j\delta_{il}\delta_{jk}$ |
| -8 | $T_j\delta_{ik}\delta_{jl}$ |
| -7 | $\delta_{il}\delta_{jk}$ |
| -6 | $\delta_{ik}\delta_{jl}$ |
| -5 | $T_{il}\delta_{jk}$ |
| -4 | $T_{jk}\delta_{il}$ |
| -3 | $T_{ik}\delta_{jl}$ |
| -2 | $T_{jl}\delta_{ik}$ |
| -1 | *copy gradient operation* |
| 0 | *no edge* |
| 1 | $T_{ijkl}$ |
| 2 | $T_{ijl}\delta_{ik}$ |
| 3 | $T_{ijk}\delta_{jl}$ |
| 4 | $T_{jk}\delta_{il}$ |
| 5 | $T_{il}\delta_{jk}$ |
| 6 | $T_{ij}\delta_{ik}\delta_{jl}$ |
| 7 | $T_{ij}\delta_{il}\delta_{jk}$ |
| 8 | $T_i\delta_{ik}\delta_{jl}$ |
| 9 | $T_i\delta_{il}\delta_{jk}$ |
| 10 | $c\delta_{ik}\delta_{jl}$ |

following examples with tensors $S_{klmn} = \delta_{km}\delta_{ln}$ and $U_{klmn} = c\delta_{km}\delta_{ln}$ :

$$\sum_{kl} T_{ij}\delta_{ik}\delta_{jl} \cdot \delta_{km}\delta_{ln} = T_{ij}\delta_{im}\delta_{jn}, \tag{7}$$

$$\sum_{kl} T_{ij}\delta_{ik}\delta_{jl} \cdot c\delta_{km}\delta_{ln} = cT_{ij}\delta_{im}\delta_{jn}. \tag{8}$$

In the first equation, we can just rename the indices of the tensor, but in the second equation we have to perform the product $cT_{ij}$, which incurs $|i| \cdot |j|$ multiplications. If two edges are multiplied with each other, we determine the resulting sparsity type of the new edge by looking up the combination in a large handcrafted table and then writing the result into the new graph representation. The new shape of the Jacobian is determined according to the rules of tensor contraction. Finally we delete the old edges from the representation as required by the vertex elimination procedure. The sparse matrix multiplication table for the computational graph representation can be found in the accompanying source code. The number of multiplications incurred by a multiplication of two edge Jacobians is computed from the sparsity types involved and the Jacobian shapes. By multiplying all values of the two Jacobian shapes with each other and masking out certain values according to the sparsity types involved, we arrive at the correct number of multiplications. For examples consider two tensors $S_{ijkl} = a_{ij}\delta_{ik}\delta_{jl}$ with sizes $(2,3,2,3)$ with sparsity type 6 and $T_{ijkl} = b_i\delta_{il}\delta_{jk}$ with shape $(2,3,2,3)$ and sparsity type 9. Thus, in the computational graph representation, we would have two non-zero entries $(6,2,3,2,3)$ and $(9,2,3,2,3)$. Then their contraction is given through

$$\sum_{kl} S_{ijkl}T_{klmn} = \sum_{kl} a_{ij}\delta_{ik}\delta_{jl} \cdot b_k\delta_{kn}\delta_{lm} = a_{ij}b_n\delta_{in}\delta_{jm}. \tag{9}$$

Then we form the product $|i| \cdot |j| \cdot |k| \cdot |l| \cdot |m| \cdot |n| = 2 \cdot 3 \cdot 2 \cdot 3 \cdot 2 \cdot 3$ where $|\cdot|$ gives the size of the dimension associated with the index. Then we use the sparsity types 7 and 9 to mask out the relevant values. In this case, we mask out $|k|$, $|l|$ and $|m|$ such that we arrive at $|i| \cdot |j| \cdot |n| = 2 \cdot 3 \cdot 3 = 18$ multiplications which is exactly the number of multiplications incurred by the dense multiplication of $a_{ij}b_n$.

# D RL Algorithm and Network Architecture

We solved VertexGame with PPO [Schulman et al., 2017] and AlphaZero [Schrittwieser et al., 2019] and used a similar architecture backbone in both cases:

- Convolutional embedding to compress the computational graph representation down using a 3x5 kernel mapped across the sequence dimension, i.e. the 3x5 filter was applied to all tokens to reduce the feature dimension. Since the computational graph representation is very sparse as each vertex typically only has a few connections to other vertices.
- Linear projection to an embedding size of 64.
- Positional encoding as described in [Vaswani et al., 2017].
- Multiple transformer layers with embedding size 64, softmax-attention, layer norm and two MLP layers with a hidden layer size of 256.

In the PPO case, we used two such backbones of with 5 and 4 transformer layers for separate policy and value networks. The policy head was an MLP of with hidden layers of sizes (256, 128) and output size 1 while the value head used an MLP with hidden layers of sizes (256, 128, 64) again with output size 1. The AlphaZero agent used the same heads on a single transformer backbone with 6 transformer layers. In all cases, the MLPs are mapped over the sequence dimension. For the policy head, this produces an unnormalized probability distribution over the actions while for the value head, we first sum all outputs to get an estimate of the input state value.

**PPO Implementation Details** We tightly followed [Huang et al., 2022] for the implementation of the PPO agent. Unless otherwise specified, all models were trained on 32 parallel environments with 4 minibatches and a clipping parameter of 0.2. We set the value and entropy weights to 1.0 and 0.01 respectively, while the learning rate was fixed to $2.5 \cdot 10^{-4}$. However, we found that the agent performed best if the rollout length was set to be equal to the number of intermediate vertices, although this is technically an implementation error.

**AlphaZero Implementation Details** The implementation is loosely based on the implementation of *Gumbel MuZero* and uses the *mctx* package provided by Google DeepMind [DeepMind et al., 2020]. However, we modified the implementation by replacing the trainable model and reward functions with our deterministic implementation of the VertexGame environment, thus effectively creating a Gumbel AlphaZero agent. Unless otherwise specified, all models were trained on 32 parallel environments with batch size 4096 and 50 MCTS simulations with 5 considered actions. We set the value and L2 weights to 10.0 and 0.01 respectively, while the learning rate was set to $1 \cdot 10^{-3}$ with a cosine learning rate annealing over 5000 episodes. For Gumbel MuZero to work properly, it is necessary to rescale the rewards so that they lie in the interval [0,1]. In the Gumbel MuZero paper, the authors designed a specific transformation that normalizes the rewards and simultaneously completes the missing values using the Gumbel distribution. The parameters of this transformation have to be carefully tuned to enable learning. In our case, we set the corresponding parameters to $c_{\text{visit}} = 25$ and $c_{\text{scale}} = 0.01$ for all cases. We trained the agent using adaptive momentum gradient-based learning [Kingma and Ba, 2014] with an initial learning rate of $10^{-3}$ and cosine learning rate scheduling over 5000 episodes on two to four NVIDIA RTX 4090 GPUs.

# E AlphaZero Results

This section contains the reward curves and elimination orders for the results presented in tables 1 and 2. Reward curves are shown for six different random seeds, namely 42, 123, 541, 1337, 1743 and 250197. The elimination orders can be directly used with the accompanying Graphax package using the *graphax.jacve* command that creates the Jacobian for a given function $f$ and a given order through *graphax.jacve(f, order=order, argnums=argnums)(\*xs)*. The orders given in tables 4 and 5 can be directly used to compute the Jacobians.

In addition to the single-graph experiments where the agent was trained only on a single task, we also experimented with training the agent on all tasks at once. For this, we randomly sampled from the 10 defined tasks to create 32 random environments. The agent was trained with the same configuration as for the single task experiments. The results are shown in figures 9 and 10 and the best achieved numbers of multiplication are displayed in table 6.

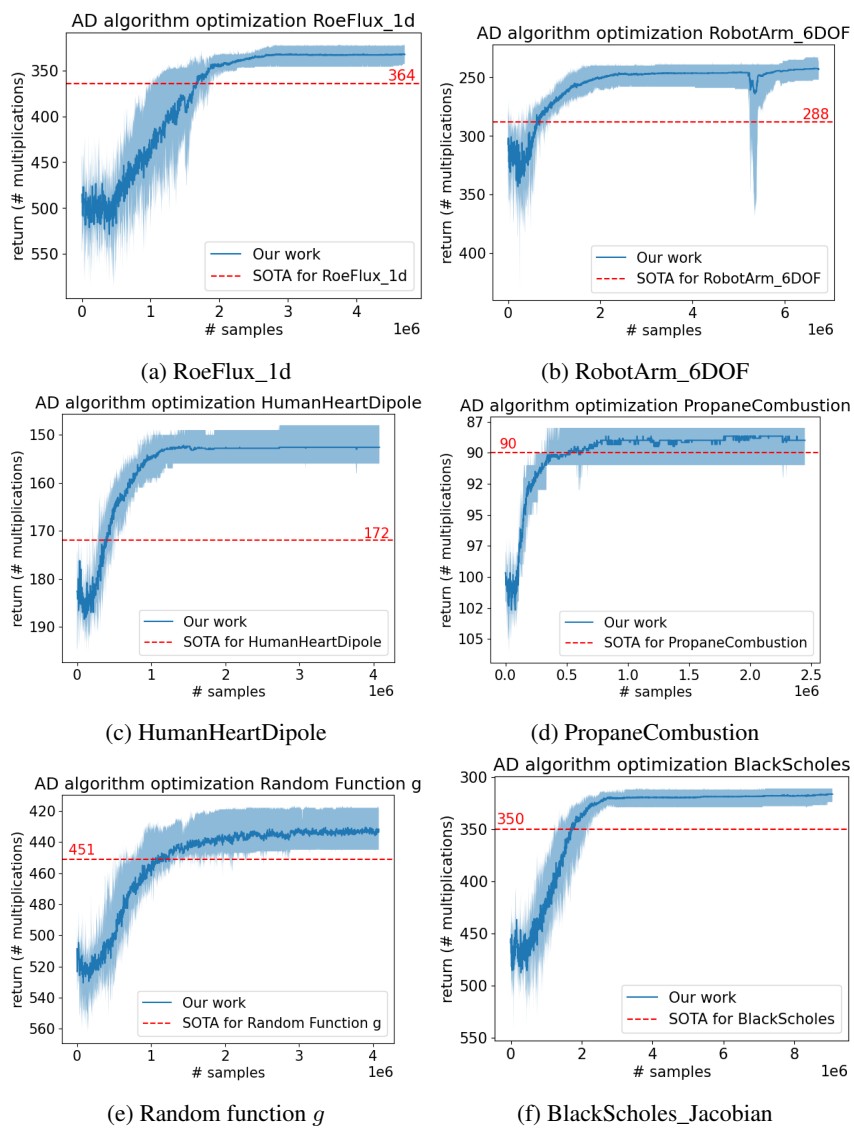

Figure 7: Learning curves of the scalar tasks for the same six random seeds with the AlphaZero-based agent. AlphaGrad manages to find better elimination orders for all tasks. The best result for each task is displayed in table 1.

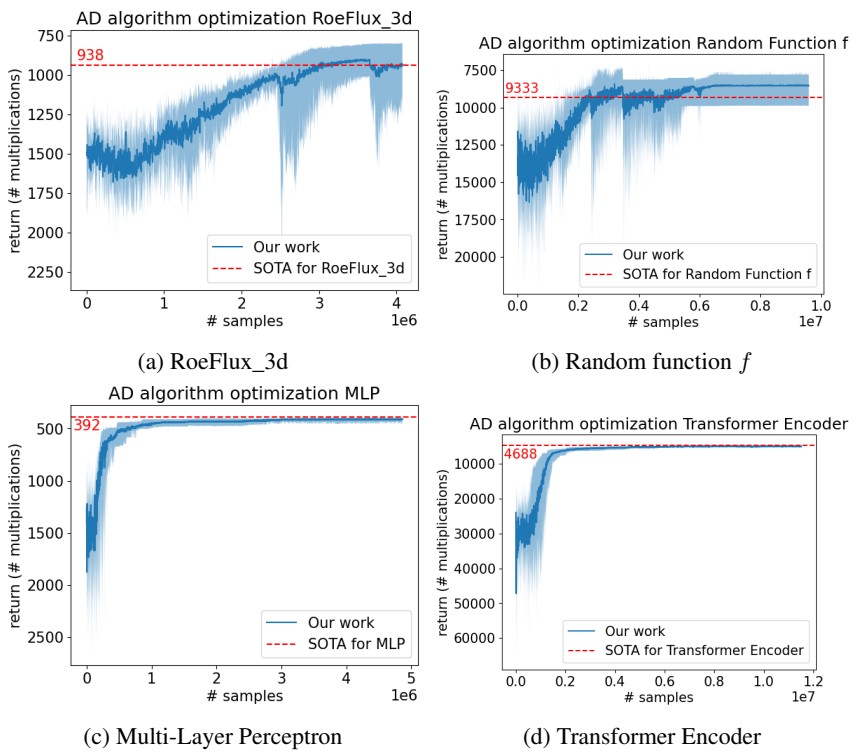

(a) RoeFlux_3d

(b) Random function $f$

(c) Multi-Layer Perceptron

(d) Transformer Encoder

Figure 8: Learning curves of the vectorized tasks for the same six random seeds for the AlphaZero-based agent. AlphaGrad manages to find better elimination orders for all tasks. The best result for each task is displayed in table 1.

Table 4: Elimination orders for the results obtained with AlphaGrad presented in table 1.

| Task | # multiplications | Elimination Order |
|------|-------------------|-------------------|
| RoeFlux_1d | 320 | [8, 82, 27, 66, 7, 78, 76, 13, 48, 42, 68, 86, 95, 4, 59, 28, 77, 54, 1, 94, 5, 58, 72, 93, 75, 31, 53, 33, 57, 90, 44, 25, 89, 88, 84, 96, 74, 92, 83, 91, 45, 51, 81, 80, 11, 10, 85, 43, 22, 73, 19, 71, 6, 18, 17, 79, 47, 50, 52, 21, 37, 38, 55, 49, 69, 35, 65, 29, 64, 16, 9, 60, 15, 61, 23, 87, 70, 67, 24, 46, 63, 39, 2, 62, 3, 41, 40, 32, 26, 34, 56, 30, 14, 98, 36, 12, 20, 100] |
| RobotArm_6DOF | 231 | [37, 18, 22, 41, 40, 8, 9, 101, 64, 36, 32, 61, 21, 14, 63, 2, 23, 82, 67, 7, 94, 15, 52, 49, 20, 97, 74, 93, 34, 77, 6, 31, 30, 104, 51, 103, 33, 105, 65, 76, 48, 45, 90, 44, 99, 95, 47, 46, 55, 73, 84, 29, 19, 79, 26, 57, 42, 43, 16, 92, 113, 112, 110, 53, 89, 35, 88, 107, 72, 70, 50, 71, 39, 83, 78, 111, 60, 58, 81, 38, 28, 5, 87, 108, 3, 91, 86, 109, 27, 54, 69, 25, 17, 106, 56, 10, 11, 75, 100, 1, 59, 98, 80, 4, 96, 13, 24, 12] |
| HumanHeartDipole | 148 | [19, 85, 11, 83, 77, 59, 81, 22, 76, 1, 9, 37, 49, 68, 69, 7, 3, 45, 51, 17, 75, 34, 66, 36, 61, 73, 71, 48, 79, 57, 40, 8, 24, 43, 39, 21, 52, 53, 16, 56, 67, 28, 42, 54, 33, 31, 30, 74, 10, 27, 47, 63, 44, 46, 6, 72, 32, 58, 55, 15, 41, 29, 13, 82, 80, 25, 26, 18, 14, 62, 5, 60, 84, 35, 64, 23, 65, 70] |
| PropaneCombustion | 88 | [12, 51, 33, 45, 10, 9, 42, 39, 1, 18, 27, 8, 61, 60, 48, 59, 36, 35, 34, 24, 26, 30, 23, 44, 43, 58, 50, 32, 40, 57, 56, 55, 54, 28, 38, 20, 21, 3, 15, 7, 2, 29, 17, 53, 5, 47, 6, 16, 14, 11, 49] |
| Random function $g$ | 417 | [1, 99, 85, 90, 87, 47, 51, 20, 3, 66, 49, 64, 13, 11, 22, 39, 61, 43, 31, 2, 6, 92, 89, 29, 16, 82, 86, 60, 24, 19, 79, 56, 63, 15, 73, 57, 50, 33, 4, 36, 70, 41, 67, 54, 30, 14, 8, 53, 78, 46, 42, 18, 17, 62, 68, 76, 65, 23, 7, 58, 38, 52, 26, 91, 34, 45, 21, 40, 35, 12, 44, 75, 25, 5, 48, 10, 59, 84, 27, 9, 71, 37, 32, 28, 74] |
| BlackScholes | 312 | [70, 16, 104, 43, 11, 15, 36, 71, 62, 42, 57, 24, 101, 74, 54, 96, 64, 65, 119, 14, 118, 50, 76, 61, 32, 19, 17, 45, 40, 59, 100, 68, 49, 126, 114, 83, 60, 116, 113, 20, 78, 25, 121, 6, 48, 31, 84, 66, 18, 28, 133, 10, 12, 58, 13, 87, 110, 29, 46, 38, 120, 92, 21, 77, 44, 107, 105, 81, 7, 56, 47, 55, 124, 67, 75, 93, 95, 79, 89, 86, 103, 82, 37, 94, 8, 52, 1, 111, 106, 23, 9, 53, 85, 90, 112, 69, 41, 34, 98, 35, 51, 22, 80, 72, 115, 91, 33, 39, 27, 99, 30, 88, 131, 123, 117, 73, 2, 109, 26, 5, 63, 128, 108, 4, 97, 102, 3, 125, 130] |

Table 5: Elimination orders for the results obtained with AlphaGrad presented in table 1 (ctd.).

| Task | # multiplications | Elimination Order |
|------|------|------|
| RoeFlux_3d | 811 | [124, 136, 56, 128, 78, 24, 1, 54, 101, 127, 121, 140, 47, 135, 67, 34, 111, 32, 100, 119, 99, 114, 125, 141, 122, 45, 65, 59, 117, 89, 116, 60, 42, 28, 74, 85, 11, 53, 36, 30, 108, 113, 55, 109, 129, 64, 91, 14, 133, 5, 10, 132, 87, 139, 110, 12, 131, 72, 8, 61, 88, 107, 6, 29, 57, 96, 118, 105, 71, 77, 112, 66, 75, 84, 143, 123, 90, 94, 137, 104, 69, 23, 22, 62, 58, 50, 130, 31, 106, 39, 48, 49, 98, 134, 93, 138, 126, 68, 115, 80, 102, 92, 79, 52, 16, 120, 95, 76, 19, 25, 73, 21, 70, 38, 35, 20, 86, 41, 4, 103, 43, 27, 3, 40, 9, 83, 13, 18, 37, 51, 46, 7, 81, 97, 63, 44, 2, 33, 82, 26, 15, 17, 145] |
| Random function $f$ | 6374 | [33, 8, 16, 77, 15, 62, 40, 58, 14, 76, 42, 60, 54, 34, 61, 72, 37, 55, 18, 75, 36, 74, 65, 26, 35, 25, 66, 38, 64, 59, 53, 20, 27, 47, 10, 69, 23, 11, 41, 79, 9, 7, 12, 63, 71, 24, 67, 51, 4, 1, 21, 3, 6, 2, 49, 13, 44, 46, 56, 17, 39, 57, 43, 32, 52, 30, 48, 31, 5, 22, 45, 19, 50, 28, 29] |
| 2-layer MLP[†] | 389 | [21, 29, 17, 15, 3, 28, 25, 30, 19, 34, 13, 8, 12, 36, 38, 31, 37, 35, 27, 33, 10, 32, 26, 20, 24, 23, 22, 18, 16, 14, 11, 9, 7, 6, 5, 4, 2, 1] |
| Encoder[†] | 4656 | [60, 81, 8, 59, 58, 41, 69, 85, 1, 46, 25, 51, 37, 17, 56, 22, 12, 75, 78, 82, 7, 66, 47, 64, 20, 88, 65, 31, 38, 6, 63, 71, 87, 19, 90, 24, 80, 83, 27, 48, 77, 49, 29, 23, 76, 9, 79, 67, 61, 26, 89, 86, 18, 34, 39, 84, 74, 70, 30, 36, 35, 72, 50, 73, 68, 62, 57, 28, 5, 55, 13, 11, 54, 53, 43, 52, 45, 44, 42, 40, 33, 32, 21, 16, 15, 14, 3, 10, 4, 2] |

Table 6: Best number of multiplications achieved in joint training mode with the AlphaZero-based agent.

| RoeFlux_1d | RobotArm_6DOF | HumanHeartDipole | PropaneCombustion | $g$ |
|---|---|---|---|---|
| 331 | 248 | 158 | n.a. | 430 |
| RoeFlux_3d | MLP | Encoder | BlackScholes_Jacobian | $f$ |
| 907 | *n.a.* | *n.a.* | 307 | 5884 |

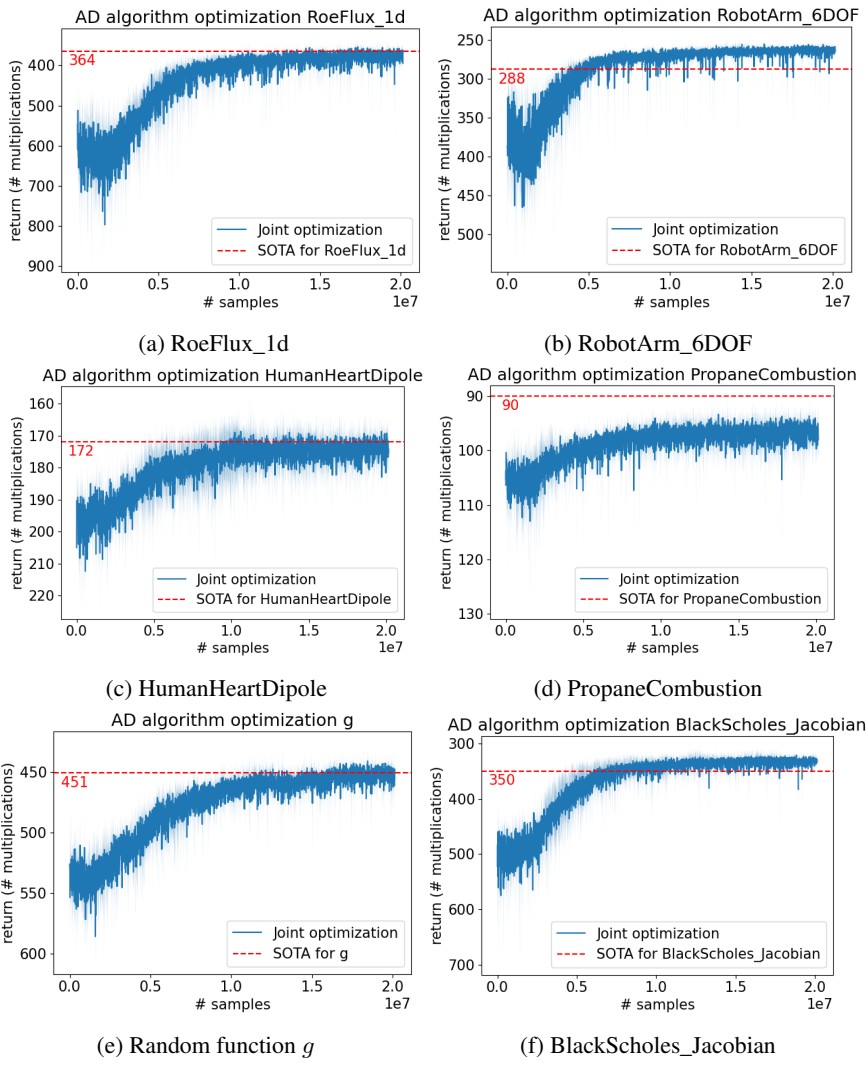

(a) RoeFlux_1d  (b) RobotArm_6DOF

(c) HumanHeartDipole  (d) PropaneCombustion

(e) Random function $g$  (f) BlackScholes_Jacobian

Figure 9: Learning curves of the scalar tasks in joint training mode for the same three random seeds with the AlphaZero-based agent. The best result for each task is displayed in table 6.

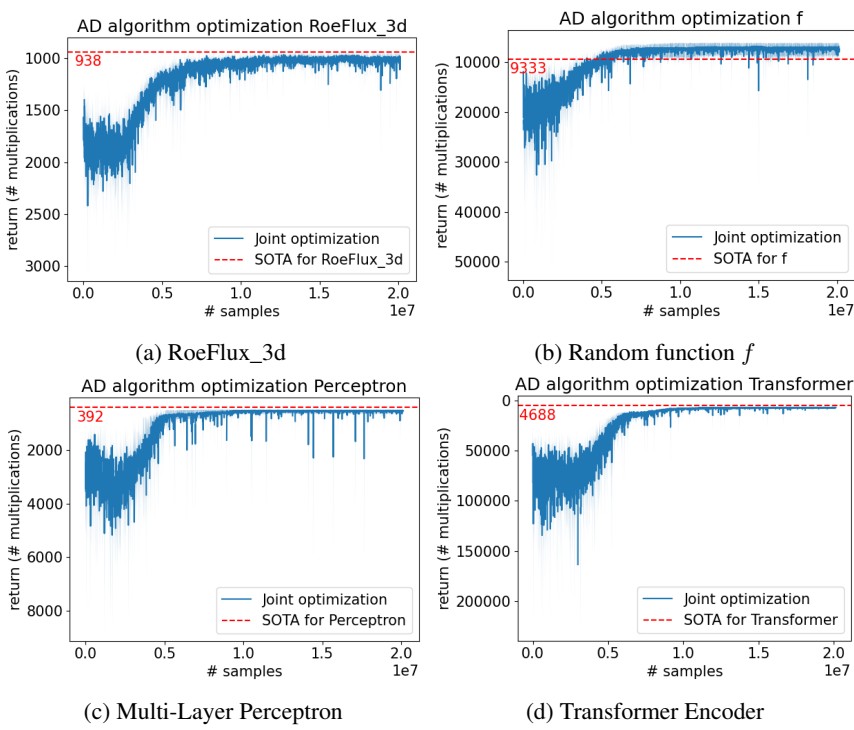

(a) RoeFlux_3d

(b) Random function $f$

(c) Multi-Layer Perceptron

(d) Transformer Encoder

Figure 10: Learning curves of the vectorized tasks in joint training mode for the same three random seeds for the AlphaZero-based agent. The best result for each task is displayed in table 6.

# F PPO Results

We ran all experiments with the configuration described in D and used the random seeds from the AlphaZero experiments. The PPO agent also manages to find better elimination orders that improve over the state-of-the-art but is outperformed by the AlphaZero agent on all tasks, sometimes by a significant margin as for example in the *RoeFlux* tasks and *random function $f$* or the *MLP* and *TransformerEncoder* where it does not find a better elimination order at all. This is to be expected since the AlphaZero agent can make use of the available model and thus select actions through planning. In other cases, the performance comes very close to the AlphaZero agent, for example in the *BlackScholes_Jacobian* or *random function $f$* tasks. Thus, the PPO-based agent might still be a viable choice because it is trained within minutes on a single NVIDIA RTX 4090 GPU, even for large tasks such as the random function $f$ and still find well-performing elimination orders. Table 7 shows the number of multiplications required by the best elimination order found by the PPO-agent. Figures 11 and 12 contain the corresponding reward curves. We did not succeed in training a joint model using the PPO agent.

Table 7: Best number of multiplications achieved by the PPO-based agent.

| RoeFlux_1d | RobotArm_6DOF | HumanHeartDipole | PropaneCombustion | $g$ |
|---|---|---|---|---|
| 324 | 245 | 162 | n.a. | 422 |
| RoeFlux_3d | MLP | Encoder | BlackScholes_Jacobian | $f$ |
| 885 | *n.a.* | *n.a.* | 313 | 6497 |

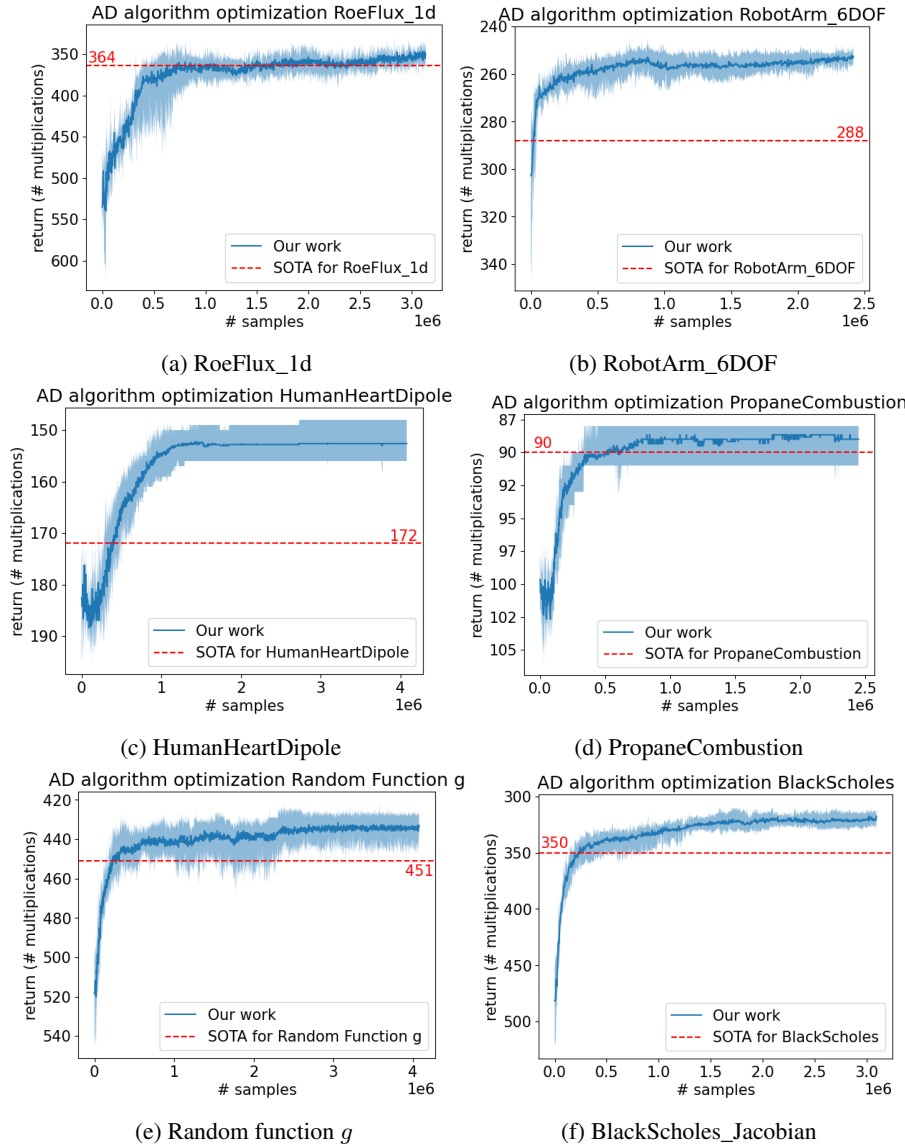

(a) RoeFlux_1d

(b) RobotArm_6DOF

(c) HumanHeartDipole

(d) PropaneCombustion

(e) Random function $g$

(f) BlackScholes_Jacobian

Figure 11: Learning curves of the scalar tasks for the same six random seeds with the PPO-based agent. It manages to find better elimination orders for many of the tasks. The best result for each task is displayed in table 7.

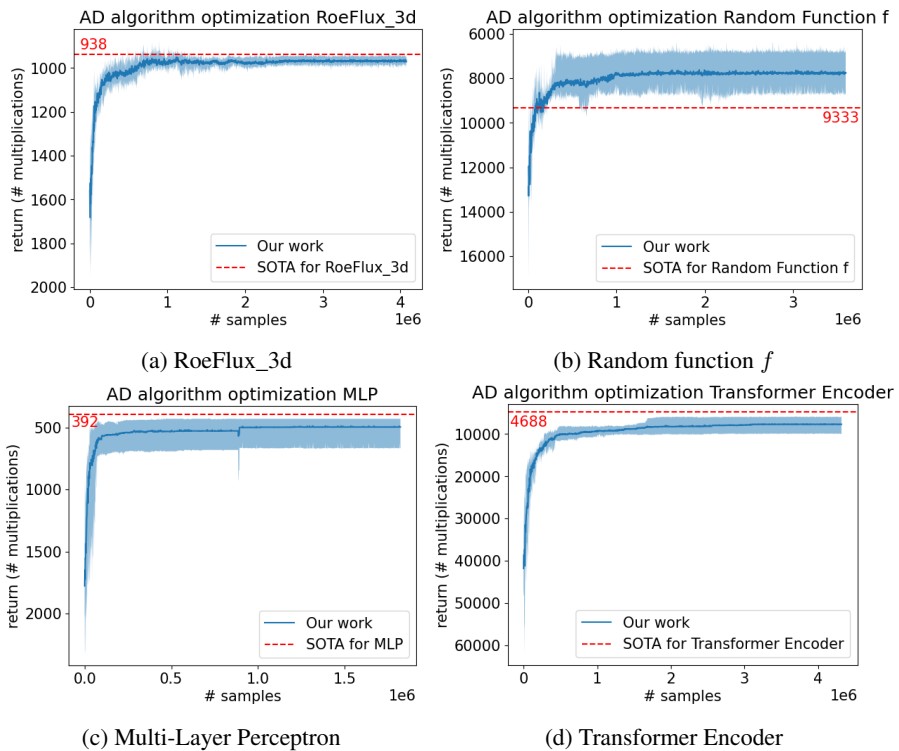

(a) RoeFlux_3d

(b) Random function $f$

(c) Multi-Layer Perceptron

(d) Transformer Encoder

Figure 12: Learning curves of the vectorized tasks for the same six random seeds for the PPO-based agent. It manages to find better elimination orders for all tasks. The best result for each task is displayed in table 7.

