# OpenReview forum: "Optimizing Automatic Differentiation with Deep Reinforcement Learning"
_NeurIPS.cc/2024/Conference — NeurIPS 2024 spotlight_

### Official Review · Reviewer_N91J · 2024-07-02

**Soundness:** 3
**Presentation:** 3
**Contribution:** 3
**Rating:** 7
**Confidence:** 3

**Summary:**

This paper studies automatic differentiation in a computational graph. The topic is classic with wide applications in scientific research, e.g., computing gradients of a neural network or Jacobians needed in numerical optimization. Classic numerical methods like standard forward- and reverse-mode differentiation (backpropagation) exist, and the novelty and contribution in this work lies in its proposal of training a reinforcement-learning (RL) agent to explore an optimized order in graph vertex elimination. The paper demonstrates the efficacy of its method by comparing the time cost of the found auto differentiation policy (measured by multiplication count and wall-clock time) with that of classic and heuristic methods.

**Strengths:**

Training an RL agent to rethink classic numerical methods is an interesting research topic. Previous examples include DeepMind’s works on matrix multiplication and sorting algorithms. Because these numerical problems are basic and fundamental building blocks in today’s scientific research, improvements in its solutions can have high potential and impact. This paper follows this trend and applies RL to another specific example: automatic differentiation. I think the paper picks a good topic and makes a meaningful contribution to the community.

The technical method in this paper also looks good to me. Using RL to explore better numerical and computing methods typically requires a careful, RL-friendly (re-)formulation of the original numerical problem, including the design of the state space, the action space, and the reward function. I think choosing to tackle vertex elimination ordering is a smart and reasonable move, which leads to a well-formulated game that captures the key structure in automatic differentiation.

**Weaknesses:**

I don’t have major concerns with the paper, but I want to mention that there are quite a few typos in the equations in the main paper. For example, I suspect “i > j” should be replaced with “i < j”, and “sum_i” in line 192 should be “sum_j”. Please double-check all equations in the paper and make sure there are no typos.

**Questions:**

I don’t have any questions for now. The key idea described in this paper is quite straightforward, and the writing in the main paper is generally good.

**Limitations:**

Admittedly, the proposed RL method does not beat classic methods across the board in the experiments. I share the similar conjecture expressed in the paper: backpropagation is a quite strong baseline for large-input, small-output computational graphs, and it can be quite difficult to beat. This may also indicate that any small improvement on this strong baseline can have a significant impact, as backpropagation is so widely used in practice.

On a related note, I appreciate that this paper is upfront about its marginal improvement over baselines in certain cases. This gives readers a balanced and well-positioned view of the proposed method.

---

> ### Author Rebuttal · Authors · 2024-08-06
>
> We thank the reviewer for pointing out the typos and will definitely fix them for the final submission.

---

> > ### Comment · Reviewer_N91J · 2024-08-10
> > **Reviewer response**
> >
> > I remain positive about this work and will maintain my score.

---

### Official Review · Reviewer_ZG2Q · 2024-07-02

**Soundness:** 4
**Presentation:** 3
**Contribution:** 3
**Rating:** 7
**Confidence:** 3

**Summary:**

The authors study the problem of computing the Jacobian of arbitrary computation graphs. The typical approach in most autodiff libraries is to implement the classic backpropagation algorithm, aggregating gradients from the end to the beginning. This is because such approaches are well suited for cases where the number of inputs is significantly larger than the number of outputs, such as neural nets.

The paper first describe alternate approaches of computing the gradient from the literature, then describe how Jacobian computation can be treated as a sequential decision making problem. The state is the current computation graph, as represented by an adjacency matrix, and the actions are choosing vertices of the computation graph to eliminate. Each vertex elimination incurs a cost of updating the partial derivatives of its neighbors, and the terminal state is when all intermediate vertices are removed and only edges between the inputs + outputs remain. The goal is to minimize the cost, which this paper measures in multiplications.

The authors try both an AlphaZero style agent and a PPO agent, and demonstrate their agents can outperform standard methods (backward AD, forward AD, etc.) on a few example functions. The empirical runtime is also analyzed, showing that the AlphaZero agent can lead to faster computation, even though the reward function does not account for all runtime factors (i.e. memory access patterns)

**Strengths:**

The paper is written well, and serves as a helpful introduction to how to optimize autodifferentiation. The paper acknowledges that it makes some simplifications to the classes of computation graphs it can be applied to (notably limits on input dimensionality make the Transformer experiment somewhat unrealistic), but demonstrate it is possible to achieve gains over typical methods using learning based methods. An analysis on real hardware is performed as well.

**Weaknesses:**

By my understanding, both the AlphaZero agent and PPO agent need to be instantiated separately for each class of functions. This is based off the note in Appendix F, that the authors did not succeed in training a joint model for the PPO agent. One of the papers cited as related work is the one for improving matrix multiplication with search algorithms, and one benefit of that work was that although the search was very specific to 4x4 matrices for example, it was possible to us the discovered results of that work elsewhere.

In comparison, I believe the current approach requires the time spent learning the PPO / AlphaZero agent to get made up by the time spent running the autodifferentiation calls. My suspicion is that the net time ends up being negative, assuming the AlphaZero setup takes longer than PPO, since the PPO setup is a few minutes of training to improve the MLP gradient computation by $$0.30 - 0.29 = 0.01$$ milliseconds  in the GPU case, which implies 18-24 million steps to hit a break-even point. (3-4 minutes / time-saved-per-step)

Still, I overall think there are interesting signs of life here, even if the method itself is not entirely practical yet.

**Questions:**

As part of the graph representation coding, there is a list of 21 different sparsity patterns to describe the intermediate edges of the computation graph. Is this list exhaustive to more general computation graphs, or are there other forms the partial derivative can take if the space of computation graphs is increased?

**Limitations:**

Yes

---

> ### Author Rebuttal · Authors · 2024-08-06
>
> We would like to thank the reviewer for the helpful input to our work.
> In particular, we agree with the reviewer that method is not entirely practical yet, but we intend to improve so that is becomes applicable to a wider range of problems.
> Automatic differentiation is an ubiquitous tool that sees application in many scientific areas. Our experiments show that there are benefits to be reaped across fields on already the simplest problems.
> Thus we are confident that the ramifications of scaling our algorithm and bringing it to practical use are huge with large potential benefits for users in all application areas.
>
> Furthermore, the reviewer's analysis made us realize that the presentation of our results does not entirely capture the actual runtime improvements achieved with our algorithm. We invite the reviewer to have a look at figure 1 in the accompanying .pdf-document where we demonstrate the scaling of our optimized algorithm for the MLP for increasing batch sizes and layer sizes. For larger MLPs and larger batch sizes, the actual runtime gains are significantly larger than what is presented in table 2 of the main text, thereby providing a stronger argument for the practical use of our method. In the words of the reviewer, with larger batch sizes, the break-even point shifts in favor of our method.
> We will adjust the main body of our submission accordingly to accommodate this.
>
> Finally, to answer the question regarding the 21 different sparsity types and their exhaustiveness:
> The list of sparsity types is exhaustive for Jacobian tensors of up to 4 dimensions i.e. $\dfrac{\mathrm{d}T_{ij}}{\mathrm{d}x_{kl}}$. Note that dimension in this case refers to the size of the mathematical objects, i.e. a number is zero-dimensional, a vector is one-dimensional and a matrix is two-dimensional etc. All computation graphs that only rely on objects that are 2-dimensional (max. 4-dim Jacobian) can be analyzed and differentiated with this procedure without any further extensions. If we want support for higher-order Jacobians, e.g. a six dimensional Jacobian tensor like $\dfrac{\mathrm{d}T_{ijkl}}{\mathrm{d}x_{mn}}$, we need to introduce new sparsity types. This is not too difficult, but for the sake of simplicity we limited ourselves to 4 dimensions as many interesting applications can already be treated in this case. In the future, we intend to extend the support for higher-dimensional Jacobian tensors to make our approach more generally applicable.

---

> > ### Comment · Reviewer_ZG2Q · 2024-08-12
> >
> > Thank you for the clarification. I will keep my score as-is.

---

### Official Review · Reviewer_tWYs · 2024-07-07

**Soundness:** 3
**Presentation:** 3
**Contribution:** 3
**Rating:** 7
**Confidence:** 3

**Summary:**

This paper considers the (combinatorially hard) optimization problem related to automatic differentiation algorithms represented via an "elimination order" based on prior work and proposes an RL formulation to solve it approximately. This follows in similar vein to recent celebrated results on matrix multiplication and sorting. The authors demonstrate that their approach is able to improve, conceptually on the first order metric of the number of multiplications as defined in the optimization search, and more practically that this can be sufficiently meaningful for runtime as they measure it on representative benchmarks , compared to the default forward/reverse mode implementations. The paper is well written and considers a reasonable variety of computational graphs in the evals. The main overhead in adopting the proposal in practice is primarily in the form of a specialized policy search for each computational graph, which factors in roughly speaking, at the compilation stage and is not reflected explicitly in the benchmarks.

**Strengths:**

- The paper is well motivated and proposes a policy search framework for a well studied combinatorial optimization problem (vertex elimination for AD).  Instead of targeting compilation (at a lower level) or dealing exclusively with optimizing individual matmuls as in prior work, the paper targets a novel alternative by focusing on vertex elimination in AD which is a ubiquitous part of all training compute graphs.
- Any non-trivial improvements on large scale computational graphs could lead to significant compute savings, and this work proposes several novel small scale benchmarks to evaluate vertex elimination algorithms which maybe of interest in its own right.

**Weaknesses:**

- The primary weakness in terms of the impact appears to be that the policy must be trained from scratch separately for each graph. This unfortunately means that there is a whole training run necessary per graph unlike all of the other competing methods (including the minimum Markowitz degree heuristic, which needs no "training", and already appears to do better than fwd/reverse mode in many situations).

- The overhead involved in running a separate optimization for every graph that is distinct from the compilation step (mostly under the hood) seems like it could makes the proposed method unwieldy for iterative usage inspite of saving some compute. Of course, any compute savings on large training graphs are valuable. However, the transformer baseline comparison indicates that the gains are more prominent in the smaller benchmarks and in the larger experiments, the reverse mode and AlphaGrad perform very close to each other.

**Questions:**

- When considering the number of multiplications, given the definition of a vertex, I assume this refers to the number of matmuls, but the costs of these vary by the dimensions of the jacobians. Do you take this into account anywhere? If not, what is the hurdle? Making the exposition a bit more explicit on the definition/implementation of the  reward is framed might be helpful.

Other comments:
- The labels for cross edges in Figure 2(d) appear to be swapped.
- Caption for Figure 2 has intermediate variable definition of $v_2=cos(v_1)$, which needs to be $sin(v_1)$ instead?
- L128 graph 2a --> Fig 2a
- In Definition 1 L138: should this be $c_{ij}=c_{jk}=0$ rather than $c_{ij}=c_{ik}=0$?
- L142, $\phi_j \circ \phi_k$ needs to be $\phi_k \circ \phi_j$ instead.
- Equation (2) needs to be $W_{ik}$ rather than $W_{ki}$?
- L324 refers to Table 2 being of theoretical value -- should this be Table 1?
- L341 s/note/not
- Figure 1 3rd box: s/executation/execution

**Limitations:**

Yes

---

> ### Author Rebuttal · Authors · 2024-08-06
>
> We would like to thank the reviewer for the valuable feedback on our work and appreciate the time the reviewer took to double-check the equations. We will implement the corrections accordingly.
>
> We partially agree with the reviewers assessment that our method requires retraining for every computational graph. While it is true that the results presented in tables 1 and 2 of the main paper were obtained with individual training, we also investigated the prospect of joint training and found that in 7 out of 10 cases our method still outperforms the baseline with a major improvement of around 10\% for the random function $f$. These results make us confident that for a larger dataset and longer training time, our method will eventually generalize out-of-distribution and predict the optimal elimination order without requiring training.
> We intend to implement this feature in future iterations of our algorithm.
>
> Furthermore, we would like to emphasize the point that this paper not only presents a novel RL-based method to optimize AD algorithms but also introduces, for the first time, a Python-based interpreter that actually allows to apply vertex elimination to a wide range of problems, including machine learning. In particular, the creation of the baselines using the minimal Markowitz heuristic are only possible due to our implementation in the first place. So while it is true that the minimal Markowitz heuristic can do better than forward-mode and reverse-mode in many cases already, it is only due to our implementation of the interpreter that this could be exploited in practice. We believe that this already makes a contribution in its own right. Furthermore, we were able to show that while the minimal Markowitz heuristic performs very well, it is outperformed by our method on all benchmark tasks, sometimes by a significant margin.
>
> Regarding the second major weakness that the reviewer pointed out, which is that our proposed method unwieldy for iterative usage, we are not entirely sure if we understand the remark correctly. Hence, we interpret it such that if the computational graph changes after the compilation stage, retraining is required.
>
> First of all, we want to clarify that the computational graph and the respective Jacobian function are independent of the input data, i.e. it is not necessary to retrain the graph every time the input of the Jacobian function changes (this might change when including control flow, which we have not considered yet).
>
> Secondly, for many practical use cases, the compute graph does not change after the compilation stage, i.e. the neural network architecture or the computational fluid dynamics equations are static. Therefore, the training phase is only necessary prior to the compilation stage once and the computed AD algorithm can be used iteratively after that for any input parameters.
> We still agree with the reviewer on the fact that the performance of our method on transformers is not satisfactory yet, but we would like to point out that theoretically there is a benefit to be harnessed as shown by table 1 in the main body of the paper. Furthermore, we deliberately included the negative outcome on the transformer case in hope of sparking future research endeavours in this direction and point out current limitations of our approach.
>
> Regarding the questions about how the reward is actually calculated and whether the shape of the Jacobian is taken into account:
> We thank the reviewer for pointing out that this part of the algorithm might not be described very well in the main body of the text and we will perform the appropriate changes to make this more explicit.
> While we gave a overview of the answer in the general rebuttal, we would like to take the time here to quickly explain in more detail how the reward is calculated:
> The reward function takes into account the shape/dimension of the Jacobians, i.e. if we have a matrix-multiplication of two dense Jacobians of shape 3x3 and 3x2, the resulting reward would be -18.
> It also takes into account the sparsity type of the Jacobian. i.e. some operations have a naturally diagonal Jacobian. In this case the matrix-multiplication becomes a pointwise operation so that for diagonal 3x3 Jacobian and a dense 3x2 Jacobian we have a reward of -6 instead. Furthermore, the *graphax* package provided together with this work is a novel interpreter that can take these sparse Jacobians into account, i.e. automatically detect these diagonal Jacobians and replace the matrix-multiplication or tensor-contraction by a pointwise multiply which is much faster in many cases. The native JAX AD package is not able to perform this kind of analysis.

---

> > ### Comment · Reviewer_tWYs · 2024-08-08
> > **Reply**
> >
> > Thanks for the rebuttal clarifications.
> >
> > > ...These results make us confident that for a larger dataset and longer training time, our method will eventually generalize out-of-distribution and predict the optimal elimination order without requiring training. We intend to implement this feature in future iterations of our algorithm.
> >
> > Amortizing the search costs across different graphs by learning to generalize between them would be nice, and the authors point out they have already seen evidence of this. However, even while considering the core results for a fixed graph, I am curious what defines distribution of the inputs that makes this evaluation distinct from that of a one-off search procedure looking for a single best elimination order not conditioned on any other inputs. Given the small sizes of the graphs involved, it would be helpful to understand the extent of learning and generalization demonstrated even if within the same graph.
> >
> > >...Furthermore, we would like to emphasize the point that this paper not only presents a novel RL-based method to optimize AD algorithms but also introduces, for the first time, a Python-based interpreter that actually allows to apply vertex elimination to a wide range of problems, including machine learning....
> >
> > I agree this is a significant and valuable contribution of the work (and distinct from the policy learning/generalization aspects). I will update my final score to take this into account.

---

> > > ### Author Response · Authors · 2024-08-11
> > > **Response**
> > >
> > > We thank the reviewer for their reconsideration and appreciate the additional questions since they are indeed quite relevant for our work.
> > >
> > > > However, even while considering the core results for a fixed graph, I am curious what defines distribution of the inputs that makes this evaluation distinct from that of a one-off search procedure looking for a single best elimination order not conditioned on any other inputs.
> > >
> > > The idea behind generalization across graphs or even within one graph is that our agent recognizes reoccurring connectivity patterns and learns how these should be optimally eliminated, conditioned on the global structure of the graph.
> > > The distribution of graphs is in that sense the distribution of all possible ways to connect $N$ vertices in a meaningful way such that they correspond to proper, mathematically well-defined and observable functions. A single computational graph and its different stages during vertex elimination do no sample this space entirely (for a single graph with 100 vertices, which is a typical size for the problems in this work, we have $100!=10^157$ possible elimination orders), thereby making it hard to find enough patterns so that the algorithm can generalize out-of-distribution without requiring further training. Also for a single graph, the patterns tend to be highly correlated, which is detrimental for generalization.
> > >
> > > There has been a lot of prior work in the AD community that aims to identify these connectivity patterns. (Griewank and Walter, 2008) contains some examples in chapters 9, 10 and 11. They call this *local preaccumulation*. In particular, they show that in many cases, an efficient solution is to first find the best elimination order locally for one or more subgraphs. These motifs can be generalized across tasks. However, in practice these recipes do not work very well because it is hard for a human to identify and generalize these for already reasonably small graphs. Given enough training samples and a reasonable search budget, we believe that our agent learns to identify these locally optimal elimination procedures and generalize them to novel computational graphs.
> > > Thus, the difference between the one-off search and the search with the model trained on a multitude of graphs is that the agent has learned more diverse wiring patterns and the corresponding elimination orders it could potentially apply.
> > > This is compounded by the fact that RL algorithms are prone to be stuck in local minima, i.e. unaware of better solutions\different connectivity patterns due to a lack of exploration. This could be partly alleviated with more compute/training time, but sometimes the one-off training on a single graph is not sufficient and the agent actually benefits from the knowledge gained from training on other graphs. This is evident when looking at the random function $f$, which is particularly hard to solve because there are no regularities and a diverse set of solutions exist. In this difficult domain, the joint training increased performance by almost 10%. We believe this is due to the agent being aware of more possible wiring patterns and their optimal elimination orders which it learned from the other tasks.
> > >
> > > > Given the small sizes of the graphs involved, it would be helpful to understand the extent of learning and generalization demonstrated even if within the same graph.
> > >
> > > This is a good recommendation for a future direction of research. To learn more about learning and generalization in our setting, it might be worth taking a look at the actual elimination orders (see Appendix C) and correlate them with the functions that were executed in the graph. This could provide some insights into some common themes that algorithm uses to optimize the AD algorithm. For example, we found that the algorithm quite often performs elimination of vertices with small Markowitz degree first, except for those that are involved in accumulation operations like summations. Those are eliminated last. More insight could possibly be gained by looking at the actual attention maps of the transformer which could give more information about which parts of the graph actually matter for optimizing the elimination orders.
> > > However, a more in-depth understanding of our method from this perspective will require additional tools which we first need to implement.

---

> > > > ### Comment · Reviewer_tWYs · 2024-08-11
> > > > **Reply**
> > > >
> > > > Thanks for the detailed response.
> > > >
> > > > My comment was specific to the current paper's evaluation in e.g. Table 1 -- it is unclear whether each row is a fixed input instance or an evaluation over a distribution of inputs. If it is the former, then the performance being evaluated is one of a pure search procedure with no explicit learning involved. Clearly, the above critique does not apply to training across a range of inputs and/or graphs.

---

> ### Author Response · Authors · 2024-08-11
> **Response**
>
> We apologize for misunderstanding the question and try to answer the questions to the best of our understanding:
>
> In the following everything is described for a single example, say RoeFlux_1d. We create an initial computational graph representation in form of the adjacency matrix. This adjacency matrix serves as an initial state for the agent similar to how for AlphaZero the initial position of the chess or Go board is fixed. In this sense, VertexGame starts from a fixed initial input for every example task, i.e every row.
>
> However, every time a vertex is eliminated, new edges are created and others are deleted and the current state of the graph and as such the adjacency matrix changes. Thus the computational graph representation evolves with every action and this evolving computational graph representation (the adjacency matrix) is fed into the agent's neural network model at every step to decide the next action. The selection of the next action is stochastic but becomes more deterministic over time since the policy, i.e. the probability distribution over actions becomes more deterministic. Thus we sample the space of possible computational graphs that can arise from the initial graph through vertex elimination with a focus on exploration early on.
> Throughout the training on a single task and its initial computational graph, the agent is thus confronted with a distribution over different graph configurations and learns the optimal action to take, similar to how in chess for different board positions the agent learns to make the next optimal move.
>
> AlphaZero learns two quantities, the policy and the value function which are realized as two distinct heads forked from the transformer backbone. The policy learns to select the next action and the value function learns to predict the value of the state, which in our case is the number of multiplications remaining until all remaining vertices are eliminated if following the current policy.
> A pure search without learning (given that we understand learning as an improvement of policy and value functions) means that those quantities would be random. The performance with random policy and value and their subsequent improvement is show in the reward curves presented in Appendix E, pages 24 and 25. We clearly see that at initialization, when the policy and value functions are random, the performance of the tree search is usually very bad and far worse than the baseline. However the performance improves over time, as evidenced by the evolving reward curves that become increasingly better until they beat the baselines. This is in our opinion clear evidence that the policy and value functions are learned over time and improve the tree search by focusing the search budget on the relevant, most promising parts of the tree. The distribution that the agent is learning on is the distribution over all possible graph configurations and their optimal next actions that can arise from performing vertex elimination on the initial computational graph of a given task example, e.g. RoeFlux_1d. Thus for each row in table 1, we start with a fixed initial input, i.e. the state at the root node in the tree is fixed. But when expanding the tree by sampling and taking actions, we create state-action-reward triplets that are sampled from the the distribution over possible graphs that can arise from the initial input graph due to vertex elimination. These samples are then used to train the policy and value functions to improve them, i.e. learn them.
>
> If we did not answer the reviewer's question, we are happy to try to clarify further!

---

> > ### Comment · Reviewer_tWYs · 2024-08-11
> > **Reply**
> >
> > Thank you for the clarification, this was very helpful.

---

### Official Review · Reviewer_ErgV · 2024-07-12

**Soundness:** 2
**Presentation:** 3
**Contribution:** 2
**Rating:** 5
**Confidence:** 3

**Summary:**

In this paper, the authors propose a novel method to optimize the number of necessary multiplications for Jacobian computation by leveraging deep reinforcement learning (RL) and a concept called cross-country elimination while still computing the exact Jacobian. The author present the search for the optimal elimination order, aimed at minimizing the required number of multiplications, as a single-player game managed by an RL agent. The results show that this approach yields up to a 33% improvement over state-of-the-art methods across various relevant tasks from different domains.

**Strengths:**

- Rephrasing the elimination order in AD as a deep reinforcement learning problem sounds somehow novel.
- The paper is well-organized.

**Weaknesses:**

- The proposed method only shows a on obvious improvement on the RoeFlux_3d and Random function. While on other benchmarks especially on deep learning related MLP and Transformer cases, the improvements are very marginal (less than 3%).
- It seems that the test cases provided are preliminary i.e., formula value evaluation instructions extracted from the whole graph, while not the the entire computational graph of a function/kernel with control-flow or mutable variables.

**Questions:**

- How does the VertexGame ensures that the reduced computational graph gives correct derivatives? Is there a grad check procedure?
- What is the application scope of the proposed method? Does it support to programs with complex control-flow (e.g., for-loop/while loop/if-condition and the combination of them)?
- The mutable variables are notorious for a AD framework as the values of different version of these variables may needed to be stored for derivatives computations during the backward pass, how does the proposed method eliminate the vertex which is a mutable variable?
- What level does the propose method works on? e.g., the source code or intermediate representation?
- Does the proposed method more close to Source Code Transformation or Tracing/Operator Overloading in a context of AD?

**Limitations:**

yes.

---

> ### Author Rebuttal · Authors · 2024-08-06
>
> We thank the reviewer for the constructive feedback of our approach and want to take the time to address some of the feedback more in-depth as compared to the general rebuttal.
>
> Firstly, we noticed that our presentation of the obtained results is not optimal and invite the reviewer to have a quick look at figure 1 in the attached .pdf-document. This figure shows the actual runtime gains obtained with our algorithm for the MLP on different batch sizes and layer sizes. It clearly demonstrates that for larger MLPs, there is a significant runtime gain to be harvested. We will make our exposition more precise to properly appreciate this result.
> Unfortunately, the transformer benchmark does not reproduce the scaling properties of the MLP, but table 1 of the main body shows that there is potential for improvement.
> In fact, we deliberately included this result in the hope of sparking future research and outlining the current limitations of our algorithm.
>
> Secondly, we partially agree with the sentiment that the results are preliminary.
> While the current benchmark tasks do not take into accounts actual kernels or control flow, many interesting practical applications do not require these features such as the MLP or the RoeFlux borrowed from computational fluid dynamics. However we appreciate the feedback and intend to include support for control flow in future iterations of the algorithm.
>
> **Q1**: How does the VertexGame ensure that the reduced computational graph gives correct derivatives?:
>
> The vertex elimination algorithm which VertexGame is based on, always yields mathematically provable exact gradient up to machine precision. This is the base assumption of our work and we thank the reviewer for pointing out that we might not have made this assumption clear enough. We will improve our manuscript accordingly. For reference, we suggest the reviewer to check (Evaluating Derivatives, Griewank and Walter, 2008, Chapters 9+10, esp. Corollary 9.2). An in-depth explanation is available in the main rebuttal.
>
> **Q2:** What is the application scope of the proposed method? Does it support to programs with complex control-flow?
>
> Our work is targeted at every field that actively applies automatic differentiation for computing Jacobians. While our method currently does not support control flow primitives such as conditionals and loops, we are eager to implement support in future iterations of our methods. One however has to make a distinction between cases where the control flow depends on variables that are to be differentiated versus variables that are not differentiated. The latter case is already implicitly supported by our package while the former case will require additional work, since . There exists an extensive body of work that addresses the more complicated case which we will harvest for future improvements.
>
> **Q3**: The mutable variables are notorious for an AD framework as the values of different version of these variables may needed to be stored for derivatives computations during the backward pass, how does the proposed method eliminate the vertex which is a mutable variable?
>
> We have considered the problems that mutable variables can cause but they are circumvented for many cases by the way JAX works. JAX leverages a functional programming paradigm which requires all functions to be pure functions with no side-effects, thereby not allowing for mutable variables that break the "self-containedness" of the pure function.
> Regarding mutable variables that do not break this, JAX guarantees referential transparency, i.e. at which point the variable has a given value by unrolling the respective computations and assigning all the intermediate values their own distinct static variables. This effectively creates a trace of the computations which can then be processed by the vertex elimination algorithm. In effect, we pay for being able to treat mutable variables with additional compute and memory consumption. It is important to note however, that in certain relevant cases such as for control flow and loops, there exist dedicated functions in JAX that avoid this costly rollout, e.g. the *jax.lax.scan* primitive. As already discussed, we intend to implement these cases in future iterations of the algorithm.
>
> **Q4**: What level does the propose method works on?
>
> Our framework works at the Jaxpression-level of JAX. Jaxpressions are a kind of intermediate representation introduced by JAX to handle many of JAX' features in a simple and elegant manner. In particular, JAX' own forward- and reverse-mode AD implementations are realized using Jaxpressions. If analyzed and transformed in the right way, the Jaxpression is actually the representation of the functions computational graph. We exploit this property in order implement vertex elimination which relies on the computational graph to work properly.
>
> **Q5**: Does the proposed method more close to Source Code Transformation or Tracing in a context of AD?
>
> Our proposed method is a mix of both, source code transformation and tracing since this is the modus operadi of JAX itself. All major features in JAX are implemented as function transformations and use a tracing feature to keep track of all operations performed. JAX introduces the Jaxpression intermediate representation to store the trace and perform the respective function transformations, including automatic differentiation. Our framework *graphax* is no exception to this. However, we want to emphasize here that to make vertex elimination work in JAX, we had to implement an entirely new interpreter from scratch. This interpreter first extracts the computational graph from the Jaxpression and then proceeds to apply the vertex elimination algorithm to obtain the optimized AD code. We consider our package *graphax* as a novel contribution in its own right because there exists no prior implementation of vertex elimination in any Python-based popular machine learning/numerical computing framework.

---

> > ### Comment · Reviewer_ErgV · 2024-08-11
> > **Response to the reply**
> >
> > Thanks for the detailed reply to my concerns and questions. I can see the performance improvements comparing to default Jax reverse mode in Figure 1 of the attached PDF. Though it does not work the same on transformer benchmark, I agree that it can be a future work as the author mention. I don't have further questions and would like to improve my score.

---

### Official Review · Reviewer_kjpN · 2024-07-12

**Soundness:** 4
**Presentation:** 4
**Contribution:** 4
**Rating:** 8
**Confidence:** 4

**Summary:**

The paper proposes to learn an efficient computation of the Jacobian of a program by reinforcement learning.
The authors base their learning procedure on cross-country elimination, a classical scheme that progressively transforms a computational graph to a bipartite graph representing the evaluation of the Jacobian. If all operations are simply scalar, a reward can simply be defined by the number multiplications accumulated in the final edges. The authors go a step further to accomodate for vector valued operations. The representation of actions and rewards can then feed to a deep reinforcement learning approach a la Alpha-zero. Experiments illsutrate the performance of the approach to find more efficient Jacobian evaluation algorithms than standard baselines on several computational graphs stemming from relevant applications. Comparisons in time are also provided to assess the practical benefits of the approach.
The authors developed a library enabling a direct use of the approach built on freely available package JAX.
Numerous comparisons are also available in the supplementary material.

**Strengths:**

- The proposed approach demonstrates benefits across tasks.
- The algorithm is developed in a package that will be released such that the method may quickly be adopted.
- The problem can serve as an interesting benchmark for deep reinforcement learning.
- The experimental details are throughly detailed to ensure preoducibility.

**Weaknesses:**

- As stated by the authors, the generalization performance of the approach may still be unclear, which may make the approach rather computationally prohibitive.

**Questions:**

- Are the performance displayed in Table 2 given after also some jit compilation?
- To be sure I understand well: currently, to compute the best elimination procedure given some function, one needs to run a full training of a deep reinforcement learning agent and then use the final policy. In terms of computational time to find the right elimination strategy, this may be rather prohibitive? (Note that showing that better elimination strategies are possible is great.)

Detail:
line 288: typo "two functions random functions"

**Limitations:**

Limitations are discussed such as the poor generalization when performing joint training and the difficulties to handle the range of rewards.

---

> ### Author Rebuttal · Authors · 2024-08-06
>
> We thank the reviewer for his constructive feedback and the careful study of our manuscript. We will correct the typos immediately.
>
> Regarding the two questions posed by the reviewer:
>
> The performances in table 2 are indeed given after JIT compilation of our AD algorithm. This was straight-forward to achieve because our package *graphax* is compatible with all other JAX transformations including JIT.
>
> The reviewer's assessment is correct as the results presented in table 2 are obtained by training on a single computational graph. However, we also performed ablation studies on joint training where we optimized the AD algorithms of all benchmark tasks simultaneously. In this case, we still observed an improvement over the baseline in 7 of 10 cases with a major improvement of almost 10\% for the random function $f$ over individual training. This makes us confident that with more data, i.e. computational graph examples and more compute we can train a model that can generalize out-of-distribution and predict optimal AD algorithms without requiring a training stage.

---

> > ### Comment · Reviewer_kjpN · 2024-08-08
> > **Acknowledging rebuttal**
> >
> > I thank the authors to answer my questions and comments.
> > I maintain my score.
> > - Applying RL methods for such a problem is an impressive feat, it even serves as a new benchmark for such approaches.
> > - Even if the results do not "significantly outperform" baselines, the fact that they can for example find better strategies than reverse-mode is quite surprising.
> > - The authors also clearly answered reviewer ErgV concerns.
> >
> > As the authors acknowledge themselves, it would be great if, for a final version, additional generalization results could be reported.

---

> > > ### Author Response · Authors · 2024-08-10
> > > **Response**
> > >
> > > We thank the reviewer for their response and will try our best to include further results in the final version.

---

### Official Review · Reviewer_PBG1 · 2024-07-14

**Soundness:** 3
**Presentation:** 3
**Contribution:** 3
**Rating:** 7
**Confidence:** 3

**Summary:**

This paper presents a novel method called AlphaGrad for optimizing computational graphs derived through automatic differentiation (AD) algorithms using deep reinforcement learning. The authors formulate the optimization of a computation graph as a single-player game where an AlphaZero agent aims to minimize the number of required multiplications. The authors demonstrate that AlphaGrad is able to improve over preexisting forward-mode AD methods in domains like computational fluid dynamics, robotics, optimization, and computational finance. Notably Deep Learning based tasks which leverage reverse-mode AD don't show significant improvements with AlphaGrad. Finally, the authors introduce Graphax, an AD interpreter for Jax that enables the optimization and execution of computational graphs discovered from elimination orders such as those proposed by AlphaGrad.

**Strengths:**

- Demonstrates a novel approach to optimizing AD by leveraging deep RL to discover optimized elimination orders.
- The paper is technically sound and provides good descriptions of these various topics for an audience who may not be very familiar with AD.
- Overall, the paper is well-written and the key ideas and results are clearly communicated.
- The work has the potential for significant practical impact as even small improvements in AD efficiency can lead to large runtime and energy savings when applied at scale.

**Weaknesses:**

- Currently, the approach has a simple model based on optimizing the number of multiplications as a proxy for runtime. As the authors note this does not capture the full complexity and ignores factors like memory access patterns.
- The baselines for this work seem somewhat limited, there are preexisting AD systems that operate on intermediate representations such as Enzyme [1] and LAGrad [2] that are able to perform much more sophisticated AD optimizations. This seems to be the future of AD and it would have been nice for AlphaGrad to operate on these IRs in a more general fashion. It would be nice if this was mentioned in the paper.
- Results don't seem to be as strong for reverse-mode AD which makes sense. Deep learning is probably the largest user of AD and any gains here could have a significant real-world impact.
- For every AD graph an RL algorithm must be run to solve for the elimination orderings. This can be quite computationally expensive and RL is known to be sensitive to hyperparameters and take quite a bit of tuning per environment.

[1] Instead of Rewriting Foreign Code for Machine Learning, Automatically Synthesize Fast Gradients. William Moses and Valentin Churavy. Neural Information Processing Systems (NeurIPS), 2020.

[2] LAGrad: Statically Optimized Differentiable Programming in MLIR. Mai Jacob Peng and Christophe Dubach.International Conference on Compiler Construction (CC), 2023.

**Questions:**

- How challenging would it be to directly optimize for metrics like execution time or memory usage rather than the number of multiplications? This seems like an important direction for making the approach more practical.
- Could you provide more details on how the random functions f and g were generated in the text?
- Were hyperparameter sweeps performed per graph? Do optimal hyperparameters for the RL agent change depending on the structure of the graph? How might we overcome this limitation?
- Why was Jax chosen instead of targeting more powerful IRs like MLIR?

**Limitations:**

The authors are forthcoming about the main limitations and highlight important areas for future work.

---

> ### Author Rebuttal · Authors · 2024-08-06
>
> We thank the reviewer for their constructive feedback, in particular for the additional references which we consider a useful contribution to shape the future of our project.
>
> We agree with the reviewer's criticism regarding the use of the theoretical number of multiplications as a proxy for actual runtime improvements. Still, we want to emphasize that this somewhat crude approach already leads to several measurable improvements.
> Thus, we are confident that when improving for actual runtime and memory access patterns, our results will only further improve. Achieving this with AlphaGrad would require an efficient hardware performance model, which we did not have available at the time of writing. We will leave this for future work, as it is promising research direction in its own right.
>
> To answer **Q1**: Implementing new reward mechanism such as runtime and memory accesses will not be too hard, since it just requires to measure the quantity of interest either directly or using one of the aforementioned hardware models. The biggest challenge lies in doing it efficiently and in a statistically robust way, since runtime and memory consumption measurements are inherently noisy. We believe that by expanding our method to incorporate methods from distributional RL, we can alleviate this issue. Given the additional additional overhead of implementing the hardware models, reward measurement mechanisms and efficient implementation thereof, we believe that it would be out of the scope of this paper.
>
> We partially agree with the statement that our baselines are limited with respect to state-of-the-art methods such as Enzyme an LAGrad. After a careful examination of both papers, we find that they operate on a different optimization level that is complementary to AlphaGrad. To prevent any misunderstandings upfront, we see this as a big advantage rather than a critique.
> LAGrad uses static optimizations to create and then optimize the differentiated code at the MLIR level. However, under the hood LAGrad is currently only able to leverage reverse-mode AD to create the derivative code. We conjecture that for an appropriate choice of function where reverse-mode AD performs worse than other AD algorithms (e.g. a function with few inputs but many outputs, which are often found in computer graphics problems such as differential rendering), the stochastic optimizations will in many cases not be able to bring reverse-mode to the same performance as other, better suited algorithms, e.g. forward-mode or minimal Markowitz. Thus, while being an amazing feat of engineering, the user is still required to select a specific AD algorithm for his use case from a very limited amount of choices. Our method instead aims to find the optimal AD algorithm, of which forward-mode, reverse-mode and minimal Markowitz are only certain instantiations.
> Thus, AlphaGrad is a tool for algorithm search while LAGrad is a tool for algorithm optimization, even though depending on the level of granularity, such a distinction sometimes might be fuzzy. Thus, we do not entirely agree that including comparisons to LAGrad in our benchmarks are fair to either side, since they operate on different levels.
> Nonetheless, we believe that combining both approaches offers a novel research direction to further extend their potential, thereby enabling even bigger gains. In particular, the handling of sparse operations seems quite intriguing.
> Regarding Enzyme, we completely agree with the reviewer that implementing AlphaGrad with a LLVM backend will surely be the more efficient choice. But due to our limited experience with LLVM and fitness to other concurrent projects in our research group, we selected JAX and its intermediate representation called a Jaxpression as an appropriate backend that enables fast prototyping (this should answer **Q4**).
> Regarding the reviewer's recommendation, we will add LAGrad and Enzyme to the introductory and discussion sections to properly appreciate their contributions and propose new research directions. In the future, we will seriously consider a reimplementation of our method using LLVM/Enzyme.
>
> To answer **Q3** regarding the brittleness of RL, we agree that the performance of many RL algorithms can massively vary which is why we selected PPO and AlphaZero as our primary agents. They are known to generalize across many different tasks with the same choice of hyperparameters. In fact, all experiments in this paper were obtained with the same set of hyperparameters and we did not find any major improvement by performing a hyperparameter search. The only deviation are the best results for the MLP and transformer cases, where we used 250 Monte-Carlo simulations instead of the default of 50. Thus we are confident, that our training method works well for many other tasks as well, leveraging a plug-and-play style mechanism to improve AD algorithms.
>
> Finally, we want to address the reviewer's **Q2** on how the random functions $f$ and $g$ were generated in more detail.
> To generate the random functions, we implemented another custom JAX interpreter which consists of a repository of elemental operations such as $\cos$, $\log$ but also matrix multiplications and array reshape operations. We then provide an interface to specify the number of input and output variables as well as the number of intermediate vertices that will be eliminated by the vertex elimination algorithm. The random function generator then randomly selects elemental functions from the repository. The number of functions that are unary, binary or perform accumulation or reshape operations can be controlled by adjusting the respective sampling probabilities.
> The function generator is part of the *alphagrad* package and can be found in the source code under *src/alphagrad/vertexgame/codegeneration/random/random\_codegenerator.py*.
> The actual implementation of the random functions can be found in the examples directory of the *graphax* package.

---

> > ### Comment · Reviewer_PBG1 · 2024-08-13
> >
> > I thank the authors for answering my questions and agreeing to update the manuscript to discuss future work on optimizing over other IRs. I want to update my score to reflect this, I think this work is important and a good first step towards applying RL for optimizing AD.

---

### Author Rebuttal · Authors · 2024-08-06

We thank reviewers for their thoughtful, detailed reviews and the additional literature suggestions. We first discuss two common themes, then address certain individual comments.

**Retraining is necessary for every function**

The best results for most of the graphs were indeed achieved with individual training. However, we also performed an ablation study where we jointly trained on all graphs at once.
The results of joint training still outperform the baseline for 7/10 tasks with even a major improvement for the random function $f$ over individual training by almost 10% (see table 1 in the attached .pdf-document). Thus, we are confident that with more computational graph examples, it is possible to train a model that eventually generalizes out-of-distribution and can predict high-performance, individually tailored elimination orders/AD algorithms for almost arbitrary functions.

**AlphaGrad does not outperform reverse-mode on ML workloads**

We acknowledge that the theoretical gain for the transformer architecture does not translate into practical runtime improvements. In fact, we chose to include this result to outline the current limitations of our approach. However, we feel that it is important to emphasize that, our algorithm, for the first time, has found a new automatic differentiation algorithm that is more efficient, i.e. requires fewer multiplications than reverse-mode/backpropagation. This shows that there is some potential to be harvested here which might spark further research since the transformer architecture is now an integral part of the ML community.
Furthermore, we invite the reviewers to have a quick look at the accompanying .pdf-document that exemplifies the scaling of the results obtained for the MLP for different workload sizes.
The relevance of MLPs for small-scale ML applications is evident and we show that for reasonably sized MLPs, we can achieve significant runtime improvements.

**Response to Reviewer #3**

**Q1**: How does the VertexGame ensure that the reduced computational graph gives correct derivatives? Is there a grad check procedure?

There exists a series of mathematical proofs that guarantee that vertex elimination always yields the correct derivatives up to machine precision. For more information, we suggest (Evaluating Derivatives, Griewank and Walter, 2008, Chapters 9+10, esp. Corollary 9.2).
Computing derivatives of a function can be seen as multiplying a long chain of Jacobians of elemental functions (such as matmuls and $
\cos$) with each other. No matter in what order they are multiplied, they will always yield the same exact Jacobian of the function. However, the order in which they are multiplied with each other severely impacts the computational performance, i.e. the number of operations that are needed to arrive at the final result. As an example take three matrices A,B,C of shapes 3x3, 3x2, 2x1. First multiplying A and B and then C requires $3\cdot 3\cdot 2 + 3 \cdot 2\cdot 1 = 24$ multiplications. Solving B and C first and then A requires $3\cdot 2\cdot 1 + 3\cdot 3\cdot 1 = 15$ multiplications. AlphaGrad tries to find the best possible order of multiplication for the chain of elemental Jacobians. The elemental Jacobians are hardcoded into the algorithm as for every other automatic differentiation tool such as Tensorflow or PyTorch. Thus, there are no approximations and vertex elimination always gives the exact gradient.

**Q2**: What is the proposed application scope? Does it support to programs with complex control-flow?

The application scope of this method includes major areas that actively utilizes automatic differentiation for gradient computation, including computational fluid dynamics, robotics and deep learning. While the current method is limited to functions without control flow, it is possible to extend the notion of vertex elimination to these cases. However, many interesting applications already arise without the need for control flow e.g. MLPs or differential kinematics of robots. Nonetheless, we agree that expanding the method to include control flow will significantly widen the scope of AlphaGrad.

**Q3**: The mutable variables are notorious for a AD framework as the values of different version of these variables may needed to be stored for derivatives computations during the backward pass, how does the proposed method eliminate the vertex which is a mutable variable?

The functional programming paradigm of JAX requires functions to be pure, i.e. to have no side effects outside the scope of the current function. This already precludes many potential problems arising from mutable variables. However, it is still possible to have mutable variables that do not violate the scope of the function. In these cases, while converting the function of interest into a Jaxpression, JAX rolls out these computations and assigns each new value of the mutable variable a new static variable (similar to static single assignment, SSA), thereby paying the price of additional compute and memory to allow for such constructs. The resulting computational graph can then be differentiated with vertex elimination.

**Q4**: What level does the propose method works on?

The method works on the Jaxpression level, which is a type of intermediate representation introduced by JAX to handle function transformations.

**Q5**: Does the proposed method more close to Source Code Transformation or Tracing/Operator Overloading in a context of AD?

The proposed method is a mixture of both approaches, operator overloading and source code transformation, since JAX also utilizes this hybrid approach.

**Response to Reviewer #4**

**Q1**: The number of multiplications is dependent on the shape of the Jacobians?

Correct. We take this into account when computing the reward and will adjust the description to make this more explicit. A full-blown explanation of how the rewards are calculated for (potentially sparse) Jacobians can be found in Appendix C.

---

### Decision · Program_Chairs · 2024-09-25

**Decision:**

Accept (spotlight)

**Comment:**

This paper is on optimizing the order of operations for Jacobian computation in a computational graph, focusing on reducing the number of multiplications required. Instead of using the standard practice of backward automatic differentiation (AD), this paper uses a reinforcement learning-based method for optimizing the automatic differentiation process.

There is a general agreement among the reviewers that the work is novel, potentially impactful, and practical. The reviewers also pointed out some weaknesses regarding computational expense for dynamic graphs and only having a modest performance improvement compared to reverse-mode AD for MLP and transformer tasks.

Overall, this is a strong paper with no major issues, and hence, I recommend accepting the paper, as do all reviewers. However, another weakness I would like the authors to address in the camera-ready paper is that the paper does not discuss a particular scope of the work. The work seems specific to static computational graphs and, hence, is more suitable for building on JAX than PyTorch. The authors mentioned in the rebuttal that “for many practical use cases, the compute graph does not change after the compilation stage, i.e. the neural network architecture or the computational fluid dynamics equations are static.” The authors should specify that and acknowledge that this work is going to be prohibitive for those using dynamically changing network architectures. The authors should also mention the implication of this issue for building on JAX vs. PyTorch in terms of implementation difficulty and overall computations.